# Testing and isolation to prevent overloaded healthcare facilities and reduce death rates in the SARS-CoV-2 pandemic in Italy

Arnab Bandyopadhyay [1,4 ✉], Marta Schips[1,4 ✉], Tanmay Mitra [1], Sahamoddin Khailaie[1],
Sebastian C. Binder [1] & Michael Meyer-Hermann [1,2,3 ✉]

## Abstract

**Background** During the first wave of COVID-19, hospital and intensive care unit beds got overwhelmed in Italy leading to an increased death burden. Based on data from Italian regions, we disentangled the impact of various factors contributing to the bottleneck situation of healthcare facilities, not well addressed in classical SEIR-like models. A particular emphasis was set on the undetected fraction (dark figure), on the dynamically changing hospital capacity, and on different testing, contact tracing, quarantine strategies.

**Methods** We first estimated the dark figure for different Italian regions. Using parameter estimates from literature and, alternatively, with parameters derived from a fit to the initial phase of COVID-19 spread, the model was optimized to fit data (infected, hospitalized, ICU, dead) published by the Italian Civil Protection.

**Results** We show that testing influenced the infection dynamics by isolation of newly detected cases and subsequent interruption of infection chains. The time-varying reproduction number ($R_t$) in high testing regions decreased to <1 earlier compared to the low testing regions. While an early test and isolate (TI) scenario resulted in up to ~31% peak reduction of hospital occupancy, the late TI scenario resulted in an overwhelmed healthcare system.

**Conclusions** An early TI strategy would have decreased the overall hospital usage drastically and, hence, death toll (~34% reduction in Lombardia) and could have mitigated the lack of healthcare facilities in the course of the pandemic, but it would not have kept the hospitalization amount within the pre-pandemic hospital limit.

### Plain language summary

Italy was heavily affected early in the COVID-19 pandemic, with healthcare facilities becoming overloaded. We use mathematical models to study COVID-19 transmission, and factors contributing to this, on a regional basis in Italy. We show that testing and lockdowns were effective in controlling disease spread. We use regional pre- and post-pandemic hospital/ICU bed occupancy to quantify the impact of the overwhelmed healthcare system upon the number of deaths. We find that increased isolation of cases could have reduced the effect of limited healthcare facilities but would not have kept hospitalizations within the pre-pandemic limit, and an improvement of hospital facilities would still have been required. We show that contact tracing and quarantine without testing could also be efficient strategies when test capacities are limited. Our findings help us to understand how to manage COVID-19 or other disease outbreaks in future.

[1] Department of Systems Immunology and Braunschweig Integrated Centre of Systems Biology (BRICS), Helmholtz Centre for Infection Research, Braunschweig, Germany. [2] Institute for Biochemistry, Biotechnology and Bioinformatics, Technische Universität Braunschweig, Braunschweig, Germany. [3] Cluster of Excellence RESIST (EXC 2155), Hannover Medical School, Hannover, Germany. [4] These authors contributed equally: Arnab Bandyopadhyay, Marta Schips. ✉email: arnab.bandyopadhyay@theoretical-biology.de; marta.schips@theoretical-biology.de; mmh@theoretical-biology.de

The COVID-19 outbreak created a worldwide pandemic causing more than 4,000,000 deaths and over 190 million total cases worldwide as of July 2021[1]. Many countries implemented non-pharmaceutical interventions (NPIs), which were effective in reducing virus spreading. This was further supported by social distancing, mask duty, and hygiene measures. Though different models of NPIs and their implementation methods have been proposed, their impact and effectiveness on disease dynamics are under scrutiny and remain a matter of global discussion[2–6]. Singapore and Hong Kong were able to contain the virus by aggressive testing[7], while South Korea adopted a trace, test, treatment strategy[8]. In a different but similarly effective approach, Japan averted the risk of contagion by isolating the whole contact clusters and by heavily relying on the self-awareness and discipline of the population[9].

The COVID-19 outbreak originated in Wuhan, People's Republic of China, in early December 2019. Within two months, it erupted and unfolded with tremendous speed in Italy, which became the European epicenter of disease spreading, forcing the government to impose a lockdown on March 9th, 2020. On March 19th, 3405 people had already died in Italy, thereby surpassing China, while 41035 people were diagnosed as COVID-19 positive. This induced Italy to shut down all non-essential businesses on March 21st. Despite the strict measures applied, in Lombardia alone a total of 28545 symptomatic people were infected by April 8th, accounting for 12976 hospital admissions, followed by Emilia Romagna (4130), Piemonte (3196), and Veneto (1839)[10]. These large numbers led to the complete collapse of the healthcare system within a few weeks of the first detection of COVID-19 cases, most notably in Lombardia where even funeral homes had been overwhelmed and were incapable of responding in a reasonable time[11]. Even though the state expanded the hospital and intensive care unit (ICU) capacities, it could not prevent the bottleneck situation of the healthcare system and presumably caused a large number of deaths for a prolonged period.

Many factors aggravated the COVID-19 situation in Italy, among which the distinct demographic structure of Italy with nearly 23% of the population of age 65 years or older[12], larger household size, and the prevalence of three-generation households compared to Germany[13] as well as limited hospital and ICU capacities. At the beginning of the pandemic, Italy focused on testing symptomatic patients only, which resulted in a large proportion of positive tests and high case fatality rates (CFR) compared to other countries[14]. A large proportion of cases remained undetected, which became a major driver of new infections. A different study estimates that in Italy the actual number of total infections was around 30-fold higher than reported, while for Germany it was less than ten-fold[15] (data up to March 17th 2020).

Compartmental models have been widely used to describe the dynamics of epidemics, for example, SIR models[16] that consider three compartments, namely susceptible, infected, and recovered, or more complex SEIR models[17,18] that take susceptible, exposed, infectious, and recovered compartments into account. Typically, these models either exclude the undetected index cases[4,17,18], or ignore their dynamic nature[19], and structurally these models are not developed to address the load on the healthcare system. Besides these epidemic models, simple algorithms exist in the literature for estimating the time-varying reproduction number and have been widely used in the context of many infectious diseases (e.g., measles, H1N1 swine flu, polio, etc.)[20]. Several studies[21,22] estimated the undetected case number in Italy, but its dynamics in the context of different testing strategies and implications on the healthcare system were not considered. Even though the general compartmental SIR and SEIR type models are useful in inferring epidemic spread and public health interventions, we needed to introduce additional compartments to investigate how the pandemic is shaped by several influential factors (e.g., dark numbers, regional testing strategies, hospital beds); for instance, we included a specific compartment for infected undetected cases ($I_X$) to analyse the impact of the region-wise undetected cases upon the evolution of the $\mathcal{R}_t$. Similarly, hospital ($H_U$ and $H_R$) and ICU ($U_D$ and $U_R$) compartments were introduced to monitor the load on the healthcare system.

Additionally, the absence of reliable symptom onset data and heterogeneity in the actual infectious period among the asymptomatic, pre-symptomatic, and symptomatic individuals require a more complex model that not only can accurately portray the dynamics of COVID-19 spread but also can disentangle the impact of intertwined factors like the variation of undetected components, limited and changing hospital and ICU availability.

Existing modeling studies that analyze the COVID-19 situation in Italy[19,23] or other regions[4,24–26], in general, did not address some fundamental aspects of the ongoing pandemic like temporal dynamics of undetected infections, the benefits of a high testing and isolation strategy, or the impact of a limited and dynamically changing healthcare capacity on the lives lost. In this study, we address the bottleneck situation of the healthcare facility, the benefits of extending hospital infrastructure, and the impact of an early testing and isolation strategy on the healthcare system with a COVID-19-specific mathematical model. To evaluate the COVID-19 situation in Italy in a realistic framework, we first estimated the undetected fraction (dark figure) of infections across different regions of Italy. We used this information to determine the parameters of the model and showed that our model is structurally identifiable. We studied the influence of the dark figure and implemented NPIs on the time-dependent reproduction number, $\mathcal{R}_t$. With data about regional hospital and ICU bed capacities, we estimated that an extra 25% of people died in Lombardia due to the overwhelmed healthcare system. We investigated the impact of early testing strategy and, alternatively, of contact tracing combined with quarantine (~10 fold more isolation of infected) policy in the setting of elevated hospital capacity as it currently stands. This strategy would have reduced the death toll by 20% to 50%.

## Methods

**SECIRD-model.** To understand the impact of potential aggravating factors, namely infections from undetected index cases, early vs late testing strategy, and limited healthcare facilities on disease progression, we developed a COVID-19-specific SECIRD-model parametrized for Italy. The SECIRD-model distinguishes healthy individuals without immune memory of COVID-19 (susceptible, $S$), infected individuals without symptoms but not yet infectious (exposed, $E$), and infected individuals without symptoms who are infectious (carrier, $C_I$, $C_R$). The carriers are distinguished into a fraction $\alpha$ of asymptomatic ($C_R$) and $(1 - \alpha)$ of pre-symptomatic infected ($C_I$). The latter are categorized into a fraction $\mu$ of detected symptomatic ($I_H$ and $I_R$) and $(1 - \mu)$ of undetected mild-symptomatic ($I_X$). Out of the $C_I$, a fraction $\rho$ gets hospitalized ($I_H$), and $(1 - \rho)$ become symptomatic but recover without hospitalization ($I_R$). Further, compartments for hospitalization ($H$) and intensive care units ($U$) were introduced to monitor the load on the healthcare system. A fraction $\vartheta$ of $H$ requires treatment in ICU ($H_U$) while a fraction $(1 - \vartheta)$ recovers from hospital without ICU treatment ($H_R$). $\delta$ and $(1 - \delta)$ represent the fraction of patients in ICU who subsequently die ($U_D$) or recover ($U_R$). The compartment ($R$) consists of patients recovered from different infection states. The *Reference Model* (Fig. 1; equations are in the Supplementary Methods 1.3) was solved with parameters in Table 1.

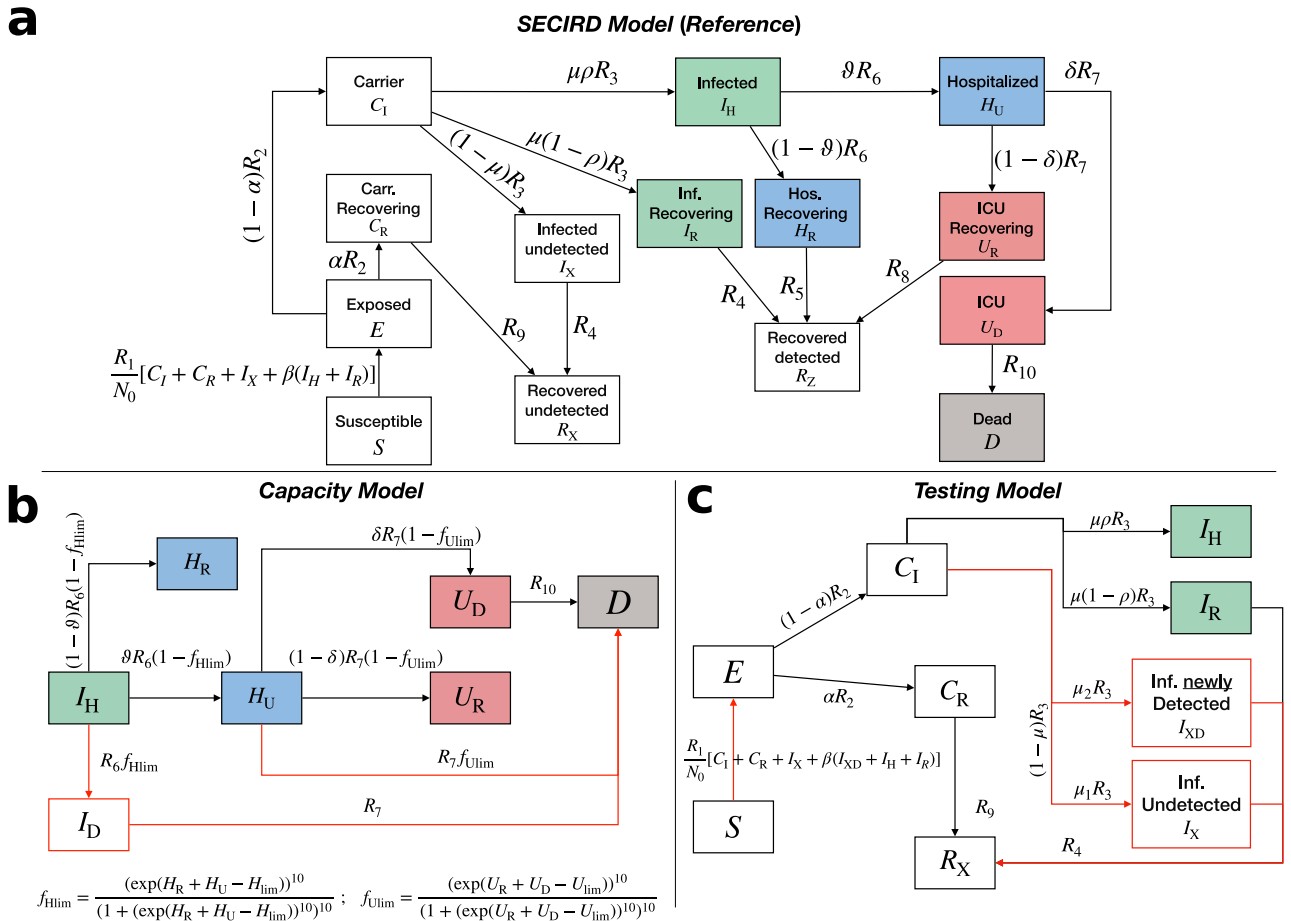

**Fig. 1 Model schemes. a** The *Reference Model* distinguishes healthy individuals with no immune memory of COVID-19 (susceptible, *S*), infected individuals without symptoms but not yet infectious (exposed, *E*), infected individuals without symptoms who are infectious (carrier, $C_{R,I}$, asymptomatic and pre-symptomatic, respectively), infected ($I_{X,H,R}$), hospitalized ($H_{U,R}$) and Intensive Care Units (ICU) ($U_{D,R}$) patients, dead (*D*) and recovered ($R_{X,Z}$), who are assumed immune against reinfection. This scheme also applies to the *Asymptomatic Model*. **b** The *Capacity Model* is a modified branch of the *Reference Model* to investigate the impact of limited hospital and ICU access onto the death toll. $f_{Hlim}$ and $f_{Ulim}$ are steep exponential functions diverting the flux from $I_H$ and $H_U$ to *D*, respectively, when hospital and ICU occupancy reached their respective current capacities $H_{lim}(t)$ and $U_{lim}(t)$. **(c)** The *Testing Model* is a modified branch of the *Reference Model* used to evaluate the impact of increasing case detection and isolation onto infection dynamics; $I_{XD}$ and $I_X$ describe newly detected and undetected cases, respectively. $R_x$ with $x \in [2,...,9]$ are per day transition rates between different states. Behavioral parameters ($\rho$, $\vartheta$, $\delta$, $R_1$ and $R_{10}$) are subject to contingent factors, like Non-Pharmaceutical Interventions (NPIs), self-awareness, availability of hospital beds, etc., and, hence, are functions of time.

**Initial condition.** Italian regions started documenting epidemiological data at different dates, at the earliest February 24th, 2020. As we considered a fixed incubation period of 5.2 days in our model, we assumed that at minimum, the first entry in the dataset (number of total cases) was the exposed number 5.2 days earlier. In addition to the documented infection, we calculated undocumented cases based on the estimated region-wise dark number by a Bayesian MCMC framework (Supplementary Methods 1.2). The sum of documented and undocumented cases was the initial exposed population. We used regional population as the initial susceptible population. For all other compartments, we started the simulation with zero. The simulation began from −5.2 days with the aforementioned initial conditions, and undetected cases were split between asymptomatic and symptomatic undetected cases according to the parameters used for those compartments.

**Parameterization.** We distinguished physiological and behavioral model parameters. Physiological parameters depend on the nature of the virus ($R_x$, $x = 2, ..., 9$) and remain unchanged throughout the analysis of the pandemic. We first determined the range of physiological values for each parameter from the literature[27,28] (Table 1) and then estimated parameters' values from a fit to the exponential growth of case numbers (infection, hospitalized, ICUs, and death cases) during the first two weeks of the pandemic. In total, 56 data points (14 daily data for those four observables) were used to estimate the physiological parameters. This initial phase was not yet affected by NPIs, public awareness, or an overwhelmed healthcare system and, thus, reflects viral properties. Some of the physiological parameters may be internally linked. For instance, hospitalization and ICU cases increase with infection cases, and disentangling those internal relations is difficult with limited data availability. It is likely that the best combination of parameters contains those internal relations. We assumed that the virus variant remained the same during the investigation period, therefore we kept physiological parameters constant throughout. However, environmental factors, testing policy, interventions, public behavior, self-isolation, hospitalization, etc. might have distorted such

**Table 1 Parameter ranges used in the *Reference Model*: determination of the boundaries for literature-based parameter set was based on the interpretation of the values given in the references[27,28,32].**

| Parameter | Comments/References | Description | Parameter ranges from literature | |
|---|---|---|---|---|
| | | | Min | Max |
| $R_1$ | Time-dependent | transmission probability of COVID-19 per each contact made with an infectious person ($C_I$, $C_R$,$I_X$, $I_R$, and $I_H$ in the model) | | |
| $R_2$ | 54,64-66 | the inverse of $R_2$ represents latent, non-infectious period following the transmission of COVID-19. 1/$R2 = 5.2 - 1/R_3$; median incubation period is 5.2 days | | |
| $R_3$ | 54,64-66 | the inverse of $R_3$ represents the pre-symptomatic infectious period | $\frac{1}{4.2}$ | $\frac{2}{5.2}$ |
| $R_4$ | 67-69 | the inverse of $R_4$ represents the infectious period for the mild symptomatic cases without requiring hospitalization (including the undetected symptomatic people ($I_X$)) | $\frac{1}{14}$ | $\frac{1}{7}$ |
| $R_5$ | 27,28,70 | the inverse of $R_5$ represents the duration for which the hospitalized cases stay in general hospital care before discharge without requiring further intensive care | $\frac{1}{16}$ | $\frac{1}{5}$ |
| $R_6$ | 28,71 | the inverse of $R_6$ represents the duration a patient stays at home before hospitalization | $\frac{1}{7}$ | 0.9 |
| $R_7$ | 28,70,71 | the inverse of $R_7$ represents the time spent in general hospital care before admission to ICU | $\frac{1}{3.5}$ | 1 |
| $R_8$ | 27,72 | the inverse of $R_8$ represents the time spent in ICU before recovery | $\frac{1}{16}$ | $\frac{1}{3}$ |
| $R_9$ | | the inverse of $R_9$ represents the duration for which the asymptomatic cases remained infectious following their latent non-infectious period | $\frac{1}{R_9} = \frac{1}{R_3} + \left(0.5 \times \frac{1}{R_4}\right)$ | |
| $R_{10}$ | Time-dependent[28,71,73] | the inverse of $R_{10}$ represents the time spent in ICU before dying | $\frac{1}{10}$ | 0.9 |
| $\alpha$ | fixed,[36-38] | undocumented asymptomatic fraction | 0.4 | 0.4 |
| $\beta$ | Assumed | the risk of infection from the registered and quarantined ($I_H+I_R$) patients | 0.05 | 0.25 |
| $\rho$ | Time-dependent | the fraction of documented infections that require hospitalization | 0.01 | 0.9 |
| $\vartheta$ | Time-dependent[74-76] | the fraction of hospitalized patients that require further intensive treatment | 0.01 | 0.7 |
| $\delta$ | Time-dependent[74-76] | the fraction of ICU patients that have fatal outcome | 0.3 | 0.9 |
| $\bar{\mu}$ | | this fraction represent the total undocumented infection including the asymptomatic cases, estimated through MLE method of the Bayesian framework | | |
| $\mu$ | | documented symptomatic fraction | $\mu = \frac{1-\bar{\mu}}{1-\alpha}$ | |

relations (e.g., the rate of increase in hospital and ICU cases as infection increases differ in different phases of the pandemic) and substantially altered the disease dynamics by impacting the transmission probability, dark number, hospitalizations, and death rate. These contingent factors affect the behavioral parameters ($\rho, \vartheta, \delta$, $R_1$, $R_{10}$). We estimated the behavioral model parameters by minimizing the sum of squared differences between the observed data (active infections, hospitalized, ICU patients, and death numbers (Italy Data on Coronavirus 2020[29])) and model simulations using Matlab's nonlinear least-squares optimizer. This procedure was repeated separately for each region in Italy in moving time windows of 7 days to account for local specifics and temporal changes in disease transmission. This moving-window technique with the size of a calendar week reduces periodic fluctuations that are an artifact of the unequal distribution of tests among the weekdays.

**Perturbation and parameter identifiability**. To generate the standard deviation for $\mathcal{R}_t$, we perturbed the behavioral parameters ($\rho, \vartheta, \delta, R_1, R_{10}$) 10% of their optimized value and sampled uniformly within this range such that the total parameter variation, $\kappa$, defined as $\log(\kappa) = \sum_{n=1}^{L} \left| \log \frac{k_n}{k_n^0} \right|$[30], remains within 10% of its reference value. $k^n$, $k_n^0$ and $L$ represent the parameters of the altered system, the reference system and the total number of parameters, respectively. We generated dynamics for 100 perturbed parameter sets for the statistical analysis.

We addressed parameter identifiability in two ways: Structural identifiability based on synthetic outbreak data and practical identifiability based on real data. In the first method, we randomly sampled parameters within a range specified in Table 1

and by using random initial conditions of model state variables. Then we used the resulting dynamics of the state variables as model observables and checked for a unique solution in the parameter space. We repeated this procedure 100 times (Supplementary Fig. 1 for a typical result). In the second method, we considered nationwide Italy data for the period February 24th to May 22nd, 2020, and fixed the physiological parameters as described in the *Parameterization* section. As we are estimating behavioral parameters ($\rho, \vartheta, \delta, R_1, R_{10}$) by considering a moving time windows of 7 days, we checked practical identifiability of these parameters in each time window. We found that the parameters are identifiable in more than 75% of the cases (Supplementary Fig. 2 for a typical results when all parameters are identifiable; and the Supplementary Methods 1.1 for more details).

**Basic reproduction number**. The basic reproduction number $\mathcal{R}_0$ is defined as the expected number of secondary infections produced by a single infection in a population where everyone (assuming no immune memory) is susceptible[31] and reflects the transmission potential of a disease. For COVID-19, the dynamics of the pandemic was influenced by several factors, like, the self-awareness in the community, interventions and policies implemented by the authorities and immunisation of the population. Therefore, the time-dependent reproduction number $\mathcal{R}(t)$ that describes the expected number of secondary cases per infected person at a given time of the epidemic, is a more practically useful quantity to understand the impact of interventions, behavioral changes, seasonal effects, etc. on the disease dynamics[20,32].

**Table 2 Estimation of the total number of infections, the Infection Rate (IR), the Infection fatality rate (IFR)[1].**

| Areas | IFR in % (95% CI) | Estimated total Infections (Undetected %) | IR in % (95% CI) | CFR in % | Detected Infections |
|---|---|---|---|---|---|
| Italy | 1.58 (1.04-1.84) | 2627807 (93.73%) | 4.37 (3.8-6.64) | 13.11 | 165155 |
| Emilia Romagna | 1.84 (1.03-2.24) | 252985 (91.69%) | 5.79 (4.84-10.22) | 13.26 | 21029 |
| Liguria | 2.08 (1.15-2.6) | 85924 (93.09%) | 5.63 (4.57-10.01) | 13.6 | 5936 |
| Lombardia | 1.66 (1.03-1.9) | 1390759 (95.53%) | 13.83 (12.16-22.19) | 18.3 | 62153 |
| Marche | 1.88 (0.88-2.47) | 58555 (90.62%) | 3.93 (3.05-8.11) | 13.56 | 5503 |
| Piemonte | 1.73 (0.78-2.12) | 258792 (92.94%) | 6.1 (5.06-13.4) | 11.05 | 18229 |
| Toscana | 1.63 (0.69-2.36) | 62671 (87.77%) | 1.43 (0.99-3) | 7.25 | 7666 |
| Valle d'Aosta | 1.54 (0.73-2.34) | 9785 (90.19%) | 9.74 (6.4-17.94) | 12.63 | 958 |
| Veneto | 1.3 (0.57-1.71) | 141466 (89.67%) | 2.77 (2.19-6.09) | 6.43 | 14624 |

[1]Based on the data provided by ISTAT up to April 15th[42,43]. Age specific IFRs are reported in Supplementary Fig. 5.

In multi-compartmental epidemic models, $\mathcal{R}_0$ can be derived with the next generation matrix method[33–35], where the Jacobian matrix consists of two factors, rate of appearance of new infections into the infection compartment ($\mathcal{F}$) and transfer of infected into other compartments ($\mathcal{V}$). The elements $G_{ij}$ of $G = \mathcal{F}\mathcal{V}^{-1}$ represent the expected number of secondary infections in compartment $i$ caused by a single infected individual of compartment $j$. The reproduction number $\mathcal{R}_0$ is given by the dominant eigenvalue of $G$ (the derivation of $\mathcal{R}_0$ is provided in the Supplementary Methods 1.4):

$$\mathcal{R}_0 = R_1 \frac{S_0}{N_0}\left[\frac{1-\alpha}{R_3} + \beta\mu\rho\frac{1-\alpha}{R_6} + \frac{\alpha}{R_9} + \beta\mu(1-\rho)\frac{1-\alpha}{R_4} + (1-\mu)\frac{1-\alpha}{R_4}\right],$$ (1)

where $N_0$ is the total population and $S_0$ is the susceptible population, both at the start of the pandemic (parameters are listed in Table 1). $\mathcal{R}_0$ was calculated using the parameter set estimated by the initial fit that considers only the first two weeks of data points as described in the *Parameterization* section. An ensemble of parameter sets (as described in the *Perturbation and parameter identifiability* section) was used to calculate the standard deviation in the $\mathcal{R}_0$.

In order to understand the impact of awareness in the population, NPIs and policies implemented by the authority upon the development of the time-varying reproduction number ($\mathcal{R}(t)$)[20], we fitted the model parameters to data in shifting time windows of one week. This approach has two advantages: first, the reproduction number $\mathcal{R}(t)$ is determined as a time-dependent variable and thus reflects the impact of NPIs on the infection dynamics; second, the moving-window dampens sudden jumps in the data because of reporting delays. In each time window, a best fit of the model parameters was found based on the cost function value (squared difference between data and simulation). In the next window, the fitting was repeated with initial conditions given by the model state in the previous time window. As described in the *Perturbation and parameter identifiability* section, an ensemble of perturbed parameter sets was used to calculate the standard deviation in the $\mathcal{R}_t$.

$\mathcal{R}(t)$ in time window $k$ reads[32]:

$$\mathcal{R}(t_k) = R_1(t_k)\frac{S(t_k)}{N(t_k)}\left[\frac{1-\alpha}{R_3} + \beta\mu\rho(t_k)\frac{1-\alpha}{R_6}\right.$$
$$\left. + \frac{\alpha}{R_9} + \beta\mu(1-\rho(t_k))\frac{1-\alpha}{R_4} + (1-\mu)\frac{1-\alpha}{R_4}\right],$$ (2)

where $\rho(t_k)$ denotes the hospitalized fraction of identified symptomatic cases in the $k$th time window. Data analysis of the clinical state of all infected cases (up to June 22nd) by the Istituto Superiore di Sanità (ISS) showed ~30% asymptomatic cases, with increasing tendency [36–38]. In a study performed in Vo' Euganeo,

Veneto, the percentage of asymptomatic cases was found to be in the range of 40%[39]. We set the asymptomatic fraction to $\alpha = 0.4$. The fraction of undetected cases $\bar{\mu}$ (*Estimation of undetected cases* section in the Supplementary Methods 1.2 and Table 2) is by definition:

$$\bar{\mu} := \alpha + (1-\mu)(1-\alpha) \Rightarrow \mu = \frac{1-\bar{\mu}}{1-\alpha}$$ (3)

*Asymptomatic Model.* In the *Asymptomatic Model*, all symptomatic cases are detected, i.e., $\mu = 1$. We compared the results from this model with those from the *Reference Model* to understand, in an ideal situation, the implication of detecting all symptomatic cases for the pandemic development.

*Testing Model.* In order to understand the influence of extra testing on infection dynamics, we adopted a model where a fraction of the undetected infected cases ($I_X$) is detected ($I_{XD}$) via testing and hence, contained. The newly detected infected ($I_{XD}$) contribute to new infections with a frequency reduced by a factor $\beta$ ($\beta < 1$) but the infectious period remains unaltered ($1/R_4$) (*Testing Model* in Figs. 1 and 2). Li et al.[40] have demonstrated, in the context of COVID-19 transmission in China, that strict control measures (travel restrictions, enhanced testing, self-quarantine, contact precautions, etc.) helped in improving the fraction of all documented infections from 14% to 65%, ~4−5 fold. For Italy, we estimated that 90% of the infections remained undetected (Table 2). We assumed that enhanced testing reduced this dark figure by daily 2% until reaching 60% (assuming similar efficiency as in China, i.e., documented infections increasing 10% to 40%). We introduced a time-dependent fraction $\mu'$ of undetected infections, which, starting from $\bar{\mu}$, was decreased daily by steps of 2% down to 60%. The asymptomatic fraction was fixed as in the *Reference Model* ($\alpha = 0.4$). The undetected portion of symptomatic is instead modified so that the fraction of undetected cases, $\mu_1(t)$, and the fraction of newly detected cases, $\mu_2(t)$, satisfies:

$$\mu_1(t) + \mu_2(t) = 1 - \mu,$$ (4)

with

$$\mu_1(t) = \frac{\mu'(t) - \alpha}{1 - \alpha}, \mu_2(t) = \frac{\bar{\mu} - \mu'(t)}{1 - \alpha}.$$ (5)

The parameters obtained by fitting the data with the *Reference Model* were transferred into the *Testing Model*. This maintains the compartmental flow of the *Reference Model* and thus ensures that the result reflects the sole effect of isolating a fraction of

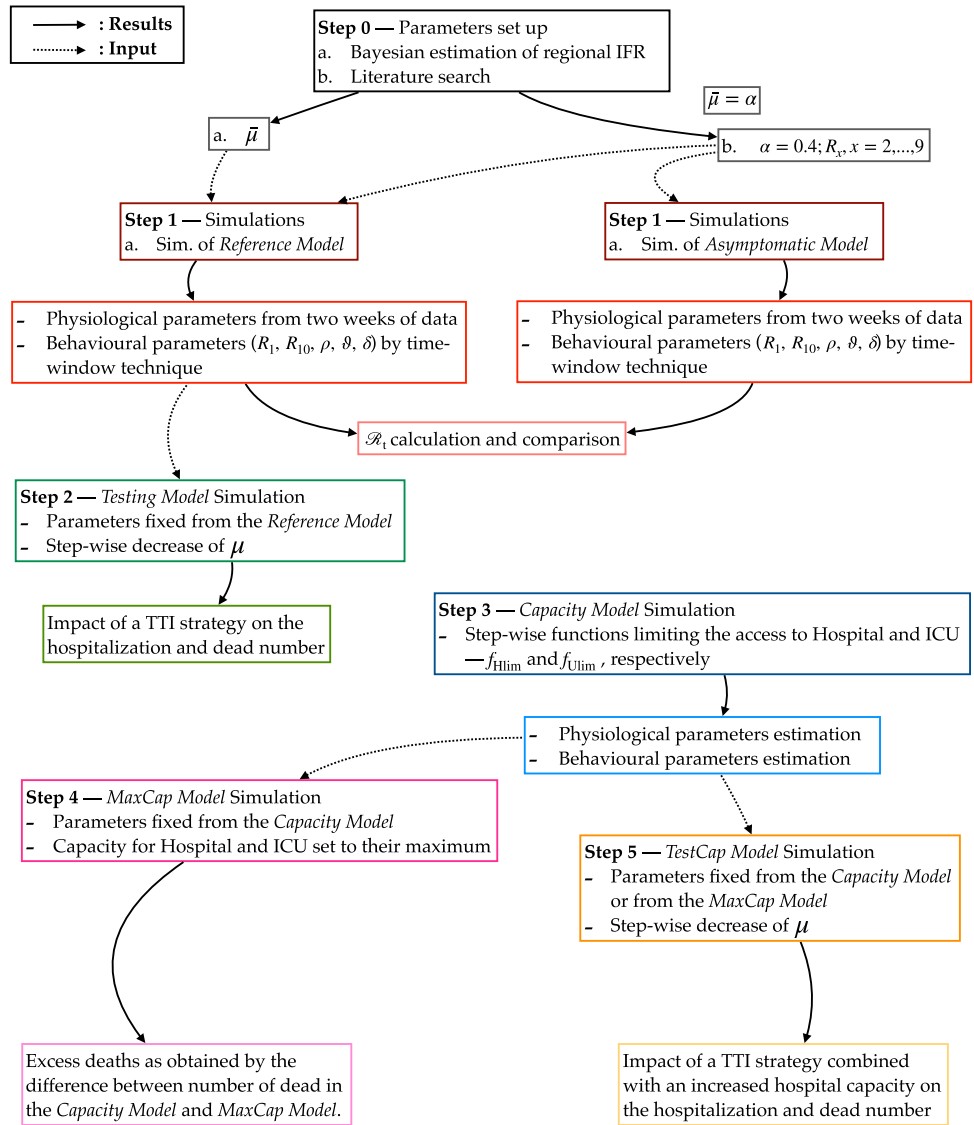

**Fig. 2 Flowchart of the study design including features and purposes of the SECIRD models.** The flowchart illustrates the steps followed to obtain the results. Each solid arrow points to the result obtained through the step from which the arrow starts, while each dotted arrow links the input to the step where that input was used.

undetected infections. Correspondingly, $\mathcal{R}(t_k)$ becomes:

$$\mathcal{R}(t_k) = R_1(t_k)\frac{S(t_k)}{N(t_k)}\left[\frac{1-\alpha}{R_3} + \beta\mu\rho(t_k)\frac{1-\alpha}{R_6} + \frac{\alpha}{R_9} + \right.$$
$$\left. +\beta\mu(1-\rho(t_k))\frac{1-\alpha}{R_4} + \beta\mu_2(t_k)\frac{1-\alpha}{R_4} + \mu_1(t_k)\frac{1-\alpha}{R_4}\right].$$
(6)

*Capacity Model.* To estimate the impact of capacity limitations of the healthcare system, we implemented time-varying capacity constraints on the hospital ($H_{\text{lim}}(t)$) and ICU ($U_{\text{lim}}(t)$) accessibility (*Capacity Model* in Figs. 1 and 2), using available data for the number of hospital and ICU beds in the different regions. Table 3 reports the pre-pandemic capacity and the increased capacity, specifically allocated to COVID-19 patients, together with the date of accomplished installation. Some regions doubled their capacity and, presumably, this extension of infrastructure has been implemented in a step-wise manner. We assumed a linear increase of the hospital ($H_{\text{lim}}(t)$) and ICU ($U_{\text{lim}}(t)$) capacity from three days before exhaustion until reaching the maximum

capacity on the date of accomplished installation. This new capacity was available thereafter. The exhaustion date was determined from the data and refers to the day at which the number of hospitalized and ICU patients became larger than the initial capacity. Before the pandemic, 85% of the hospital beds and 50% of the ICU beds were occupied[41]. In the *Capacity Model*, 15% and 50% of the pre-pandemic total capacity (Table 3) was considered as the baseline capacity of hospital and ICU beds, i.e., the starting values of $H_{\text{lim}}(t)$ and $U_{\text{lim}}(t)$, respectively.

Upon reaching the capacity limit, the influx should be stopped until a vacant bed is available. This could be achieved by introducing a Heaviside step function or any other piecewise method, but this type of function introduces discontinuities and makes solving the ODEs computationally demanding and error-prone. Here, we introduced two functions ($f_{Hlim}$ and $f_{Ulim}$) that behave like a step function but are continuous in nature. $f_{Hlim}$ and $f_{Ulim}$ return 1 as long as there are free hospital or ICU beds and 0 otherwise. In the *Capacity Model*, we introduced the compartment $I_D$, to which the flux from $I_H$ is directed when the hospital capacity is reached. Then, in a sharp transition, patient access to

**Table 3 Hospital bed and ICU capacity before and in the course of the pandemic[41,77][1,2].**

| Regions | ICU | Beds | Added ICU | Date ICU | Added beds | Date beds |
|---------|-----|------|-----------|----------|------------|-----------|
| Abruzzo | 109 | 4410 | 67 | 31/03/2020 | 537 | 23/04/2020 |
| Basilicata | 49 | 1861 | 24 | 17/03/2020 | 139 | 17/03/2020 |
| Calabria | 153 | 5739 | 60 | 11/04/2020 | 126 | 11/04/2020 |
| Campania | 506 | 17977 | 104 | 11/04/2020 | 773 | 14/04/2020 |
| Emilia Romagna | 449 | 17295 | 259 | 24/03/2020 | 2189 | 24/03/2020 |
| Friuli Venezia Giulia | 127 | 4333 | 102 | 02/04/2020 | 358 | 08/05/2020 |
| Lazio | 557 | 20817 | 323 | 24/04/2020 | 1527 | 21/04/2020 |
| Liguria | 186 | 5690 | 127 | 07/04/2020 | 1241 | 01/04/2020 |
| Lombardia | 859 | 37767 | 939 | 03/04/2020 | 11673 | 12/04/2020 |
| Marche | 115 | 5183 | 132 | 31/03/2020 | 638 | 06/04/2020 |
| Molise | 31 | 1225 | 12 | 28/03/2020 | 31 | 07/04/2020 |
| Piemonte | 317 | 16313 | 500 | 08/03/2020 | 4451 | 16/04/2020 |
| Puglia | 302 | 12531 | 297 | 11/04/2020 | 1027 | 26/04/2020 |
| Sardegna | 123 | 5739 | 40 | 14/04/2020 | 92 | 07/04/2020 |
| Sicilia | 392 | 15821 | 312 | 23/04/2020 | 1632 | 04/05/2020 |
| Toscana | 377 | 12021 | 247 | 06/04/2020 | 1350 | 05/04/2020 |
| Umbria | 70 | 3259 | 35 | 25/03/2020 | 131 | 11/04/2020 |
| Valle d'Aosta | 12 | 481 | 25 | 03/04/2020 | 262 | 03/04/2020 |
| Veneto | 487 | 17512 | 331 | 17/03/2020 | 1910 | 17/03/2020 |
| Bolzano (AP)[2] | 40 | 2047 | 66 | 16/04/2020 | 442 | 03/04/2020 |
| Trento (AP)[2] | 32 | 2113 | 70 | 02/04/2020 | 382 | 07/04/2020 |

[1]*ICU* and normal *Beds* represent the pre-pandemic total beds. In the simulation we used 50% of ICU and 15% of normal beds as baseline capacity. *Added ICU* and *Added beds* represent increased allocation specifically for COVID-19 patients. *Date ICU* and *Date beds* is the date when the additional beds and ICUs were in place.
[2]*AP* autonomous province.

the hospital or ICU is reduced by the fractions $f_{Hlim}(t)$ and $f_{Ulim}(t)$, respectively:

$$f_{Hlim} = \frac{\left(\exp\left(H_R + H_U - H_{lim}\right)\right)^{10}}{\left(1 + \left(\exp\left(H_R + H_U - H_{lim}\right)\right)^{10}\right)^{10}} \quad (7)$$

$$f_{Ulim} = \frac{\left(\exp\left(U_R + U_D - U_{lim}\right)\right)^{10}}{\left(1 + \left(\exp\left(U_R + U_D - U_{lim}\right)\right)^{10}\right)^{10}}. \quad (8)$$

Both factors increase fatal outcomes of infections when hospital and ICU capacities are reached. We assumed that inaccessibility of hospital or ICU leads to faster and more frequent death. In particular, when the ICU capacity is reached, people in the hospital compartment ($H_U$) die after $1/R_7$ days which is faster than via the hospital-ICU-dead route ($(1/R_7 + 1/R_{10})$). Similarly, when the hospital capacity is reached, people in the infected compartment ($I_H$) die after $1/R_6 + 1/R_7$ days, satisfying $1/R_6 < 1/R_6 + 1/R_7 < 1/R_6 + 1/R_7 + 1/R_{10}$ (see *Reference Model* in the Supplementary Methods 1.3).

The *MaxCap Model* is defined by the same equations as the *Capacity Model*, but the parameters, $H_{lim}$ and $U_{lim}$ were set to the maximum hospital and ICU capacity, respectively, from the beginning of the simulations.

**Data and code**. Italy COVID-19 data of infected cases, hospitalized and ICU patients, and death numbers were provided by the *Protezione Civile Italiana*[29]. Demographic and mortality data used to estimate IFR, are available from the *Italian Institute of Statistics* (ISTAT) website[42,43]. ISTAT collects mortality data from the Italian *National register office for the resident population* (ANPR). An automated method was implemented, and parameter estimation was carried out in Matlab 2019b[44] with a combination of the Data2Dynamics framework[45]. The code is available at https://github.com/arnabbandyopadhyay/COVID-19-in-Italy, and has been archived on Zenodo at ref. [46]. For the

Bayesian estimation of COVID-19 IFR of Italian regions, see Supplementary Methods 1.2.

**Reporting summary**. Further information on research design is available in the Nature Research Reporting Summary linked to this article.

## Results

**Region-wise infection fatality rate (IFR)**. To estimate undetected infections amid the COVID-19 pandemic, we analyzed the mortality rate of previous years and the deaths in 2020. Demographic and death data of the Italian regions have been collected from the Italian Institute of Statistics (ISTAT). The observed mortality of 2020 was substantially higher than in previous years in those Italian regions where the pandemic started—e.g., Lombardia, Veneto, Piemonte (Supplementary Fig. 3). We estimated the total number of infections, including undetected cases, and the associated IFR (defined as the percentage of deaths among all infections, including the undiagnosed infections) for each region with a Bayesian framework by adapting a standard binomial model (Supplementary Methods 1.2, Supplementary Fig. 4). Though literature is available on IFR estimates in Italy, the impact of the undetected infections on the healthcare facilities and infection dynamics, considering different testing strategies adopted by the regions, were not sufficiently addressed[21,22].

The regional IFR, the estimated total infections, Infection rate (IR), and the undetected fractions were summarised in Table 2. In the northern regions where the outbreak occurred first—e.g., Emilia Romagna, Piemonte, and Veneto—the undetected infections were nearly 10-fold more than the reported cases, and in Lombardia, it was more than 21 fold. We observed substantial heterogeneity of the IFR across different age groups (Supplementary Fig. 5). For ages below 60, it was as low as 0.05%. The IFR was drastically higher in the 81+ age group (9.5% to 20%, Supplementary Fig. 5). Despite Italy having the highest COVID-19 deaths in Europe, the estimated infection rates (IR) were relatively low (highest in Lombardia ~13%[12.16−22.19%, 95%

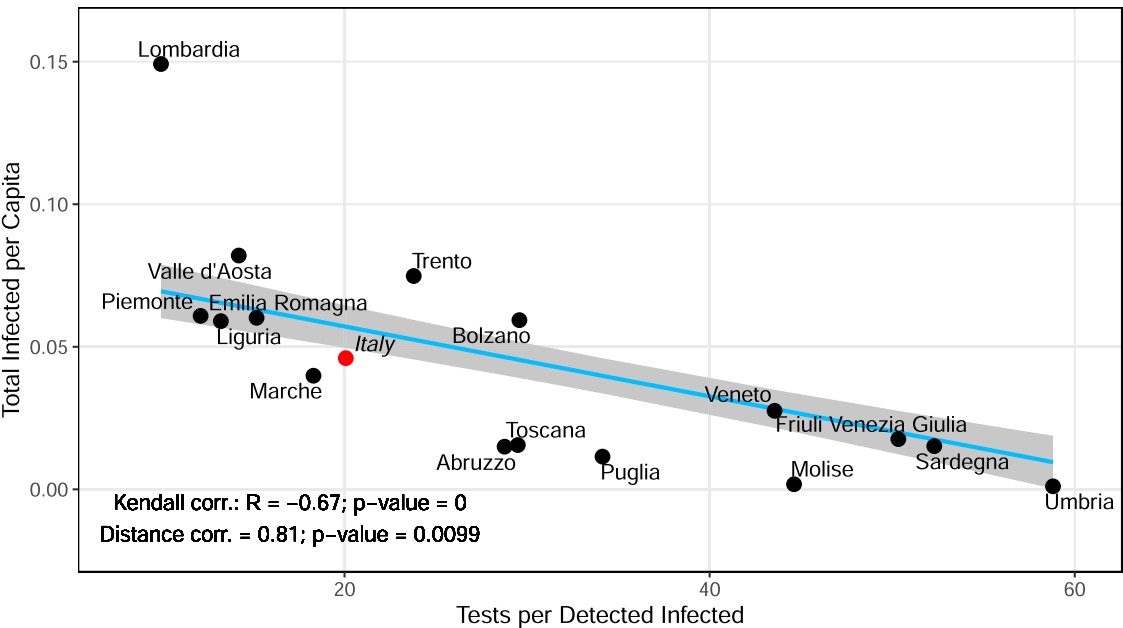

**Fig. 3 Impact of tests on the total infections.** Kendall and distance correlation between the number of tests performed per infection and total infections per capita. By considering 16 regions and Italy, a significant negative correlation confirms that the infection dynamics can be controlled by aggressive testing, which is further supported by the infection epidemic curve in Fig. 4a. Data are considered up to April 15th. Light blue line: linear regression fit; gray shaded area: standard error; black dots: region-specific values; red dot: nationwide value; *R*: correlation coefficient; *p*: significance.

CI]) across all regions, and hence the population was far from reaching the herd immunity threshold (~70% with formula $1 - \frac{1}{\mathcal{R}_0}$, assuming no previous immune memory and considering the lowest mean of reported $\mathcal{R}_0$ for Covid-19 is ~3)[47–49]. Our estimated total infections (detected + undetected), IR, and IFR are close to the Imperial College London report 20 on Italy[22].

During the early outbreak, different northern regions in Italy adopted different testing strategies, which heavily influenced the infection dynamics. Lombardia and Piemonte followed the World Health Organization (WHO) and central health authority recommendations by mainly testing symptomatic cases, while Veneto implemented a more extensive population testing. Lombardia (~10 million inhabitants) has suffered around 14,000 deaths by the end of April, which is more than half of all COVID-19 deaths in Italy. In comparison, Veneto, with a population of 5 million, has suffered around 1,400 deaths. This difference is reflected in the CFR and IR values (Table 2): respectively, 6.43% and 2.77% in Veneto, while 18.30% and 13.83% in Lombardia, despite the geographical proximity.

To understand whether implementing different testing strategies succeeded in keeping undetected and the overall infection amount under control, we investigated the association between the tests performed by regions with the total number of infections (detected and undetected, Table 2) in the early phase of the pandemic. According to the null hypothesis, the total number of infections would be uncorrelated with the testing volume, as testing only discovers undetected infections and therefore should not impact the total infection. This hypothesis holds when testing does not influence the infection dynamics. We measured the Kendall and distance correlation between total infection per capita and total tests performed (up to April 15th, 2020) per reported infection. This yielded a significant correlation in both tests (Fig. 3, Supplementary Fig. 6). The negative correlation indicates that testing influenced the infection dynamics by isolation of newly detected cases and subsequent interruption of infection chains (similar results obtained with CFR/IFR and Tests per detected infected, Supplementary Fig. 6). The impact of

testing is further supported by the observed infection dynamics. Regions with intense testing, like Veneto and Toscana, flattened the infection curve by the middle of April, while for Lombardia, Liguria, and Piemonte with inadequate testing, this was delayed by three weeks (first week of May 2020, Fig. 4a).

**Influence of undetected cases on $\mathcal{R}_t$.** To understand the influence of undetected cases and installed NPIs on infection dynamics across different regions in Italy, we used the COVID-19-specific *Reference Model* to explain the dynamics of infected, hospitalized, ICUs, and death numbers provided by the *Protezione Civile Italiana*[29] (Fig. 4a, Supplementary Fig. 7). As described in the *Parametrization* section, based on the ensemble of parameters estimated by using the first two weeks of data, we calculated the basic reproduction number $\mathcal{R}_0$ according to equation (1). Estimated behavioral parameters in every window (keeping the physiological parameters constant) were used to calculate the reproduction number $\mathcal{R}_t$ (Fig. 4b, Supplementary Fig. 8) according to equation (2). The sudden increase in reported cases resulted in an overshoot in the $\mathcal{R}_t$ curve at the beginning. Due to the nationwide NPIs, increased public awareness inducing self-isolation and social distancing, the reproduction number continuously decreased, approaching unity at the end of April. In the regions with many undetected infections, like Emilia Romagna, Lombardia and Piemonte, the reproduction number reached unity in the first week of May, while in Veneto and Toscana it reached unity in the middle of April and was substantially lower by the first week of May. As $\mathcal{R}_t$ functionally depends on many factors (see $\mathcal{R}_t$ formula in the Methods section), we opted for a sensitivity analysis to find the important factors that regulate $\mathcal{R}_t$. It revealed that $\mathcal{R}_t$ is highly sensitive to the transmission probability $R_1$ and the dark figure $\bar{\mu}$ (Supplementary Fig. 9). The influence of installed NPIs, social distancing, awareness etc. are embedded within $R_1$ and therefore, their impact is reflected in the decreasing trend of $\mathcal{R}_t$ in all regions. The benefits of testing and the impact of undetected cases on the $\mathcal{R}_t$ evolution can as well be inferred by comparison of $\mathcal{R}_t$ in the

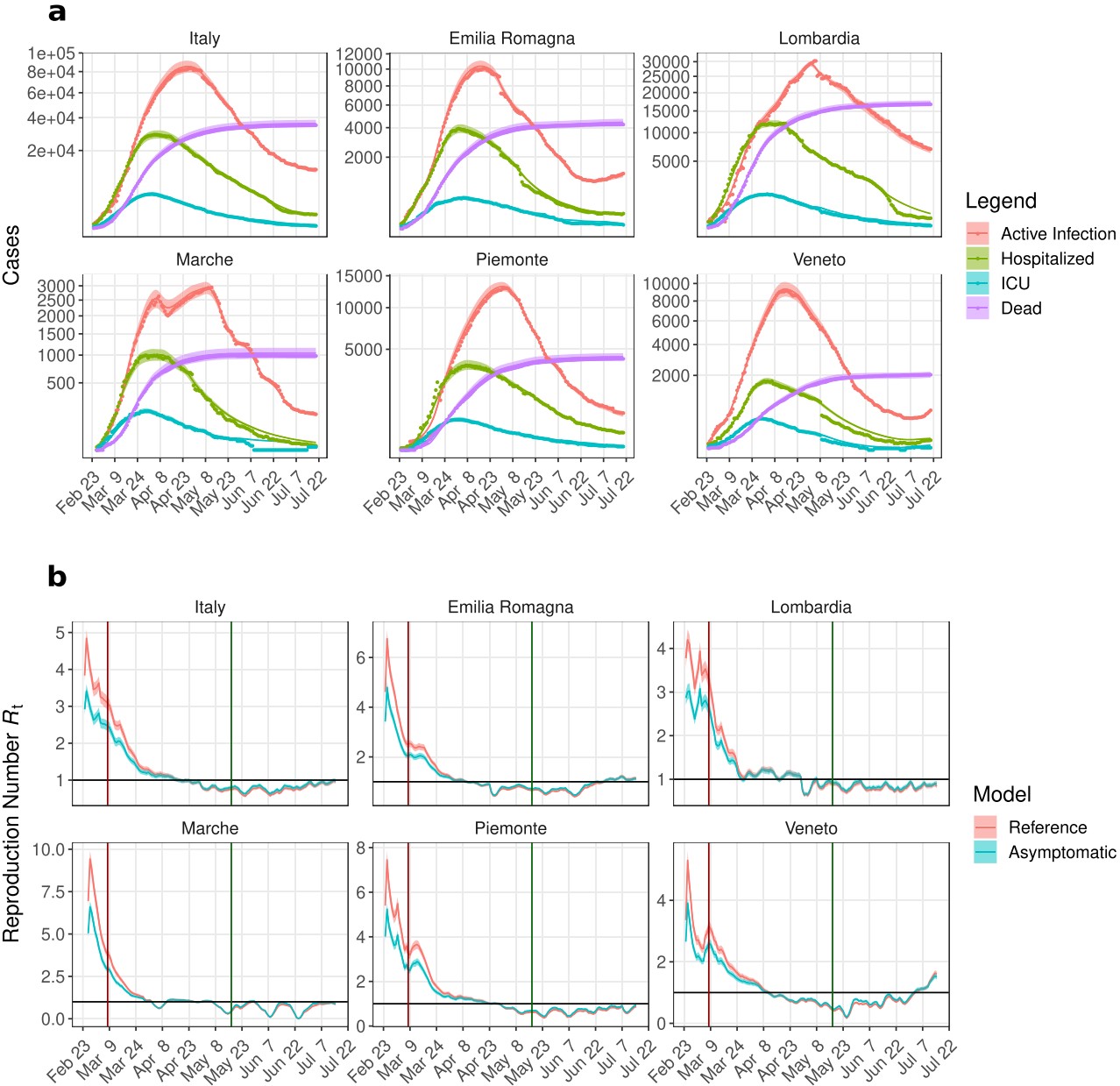

**Fig. 4 *Reference Model* and time-dependent reproduction number $\mathcal{R}_t$ in different regions of Italy. a** Active infections, hospitalized, Intensive Care Units (ICU) and death data (nationwide and region-wise) were fitted in a sliding one week time window. Parameter ranges listed in Table 1 were used and the behavioral parameters ($R_1$, $R_{10}$, $\rho$, $\vartheta$, $\delta$) were estimated in each time window (see Methods); dots: data; continuous lines and shaded area: respectively, mean and standard deviation of all dynamics generated by using 100 perturbed parameter sets (see Methods). **b** Dynamics of the time-dependent reproduction-number, $\mathcal{R}_t$, resulting from the fit with the *Reference* (red) and the *Asymptomatic Model* (blue). Statistics of $\mathcal{R}_t$ were obtained by fitting the data with 100 perturbed parameter sets; continuous lines and shaded area are, respectively, mean and standard deviation. Vertical lines correspond to the Lockdown implementation (dark red) and release (dark green). Black horizontal line represents $\mathcal{R}_t = 1$.

*Reference* and the *Asymptomatic Model* with less undetected cases ($\bar{\mu} = \alpha$, $\mu = 1$).

In the *Asymptomatic Model*, all symptomatic infections are detected and, hence, this reduces secondary infections ($\beta \ll 1$). Removal of the highly infectious compartment ($I_X$), causes a lower turnover from susceptible to exposed. This effect is apparent in the initial phase of the pandemic when $\mathcal{R}_t$ in the *Asymptomatic Model* is much lower than in the *Reference Model* (iris-blue and red line, respectively, in Fig. 4b). In the long term, the $\mathcal{R}_t$ curves from the two models converge, despite the fraction of undetected cases being constant throughout the simulations: $\bar{\mu}$

in the *Reference Model* and $\alpha$ in the *Asymptomatic Model*. This is consistent with the effect of the restriction measures, limiting the spreading through undetected cases. The difference in the $\mathcal{R}_t$ curves of these two models is larger in the pre-lockdown period. As soon as people's behavior has started changing either by self-awareness or by imposed restrictions, the turnover from susceptible to exposed is reduced. This caused the adjustment of $R_1$ and the merging of $\mathcal{R}_t$ in both models. Thus, the influence of undetected infections on $\mathcal{R}_t$ wanes (Fig. 4b) because of two reasons, first, restricted contacts due to the nationwide lockdown, and second, the enhanced testing strategy adopted by the regions.

**Increased Test and Isolate (TI) strategy to reduce hospitalizations**. As the impact of undetected cases on $\mathcal{R}_t$ faded with time, we wanted to quantify the benefit of an increased test and isolate (TI) strategy by implementing an early (from March 2nd, 2020, i.e., one week before lockdown) and late (from March 15th, 2020) testing strategy. Many countries have implemented a Test, Trace and Isolate (TTI) strategy that has a high detection rate, and several modeling studies indicate that a high proportion of cases would need to isolate to control the pandemic[50,51]. In our case, an increased test and isolate strategy with a high success rate of detection and isolation (as mentioned in the Methods section) is a phenomenological implementation of a TTI strategy. In the *Testing Model*, we assumed that a fraction ($\mu_2(t)$) of the symptomatic undetected cases, $I_X$, was detected by tests ($I_{XD}$) and, hence, became less infectious ($\beta \ll 1$, Methods, Fig. 1). We maintained the compartmental flow from the *Reference Model* by using the same parameter set in order to determine the impact of isolating undetected infections by targeted testing and home quarantine of contact clusters around identified infections. This *in silico* experiment resulted in a substantial increase in the number of detected infections but reduced the number of required hospitalizations. The early TI scenario resulted in up to ~32% peak reduction of hospital bed occupancy, which reduced death numbers by up to ~44% (Fig. 5a, b) depending on the region. The late TI scenario resulted in a situation similar to reality, namely, an overwhelmed healthcare system with little decrease in peak hospital occupancy. However, late TI still reduced death numbers by up to ~24%. Although enhanced testing increased the total number of detected infections in comparison to the real number of detected cases, $\mathcal{R}_t$ fell and reached unity three weeks earlier (Supplementary Fig. 10). In line with previous results[50], this result suggests that TTI strategies are efficient in decreasing disease propagation. South Korea successfully mounted a targeted testing strategy to contain disease spreading without imposing strict measures, like lockdowns or immigration control[8].

Having established that increased detection and isolation lower hospitalization rates, we investigated whether the same relation can be inferred from the data directly. We calculated the correlation between the median hospital occupancy and the total number of tests performed per infected up to May 22nd, 2020 (Fig. 5c). Regions with low testing were associated with the healthcare system hitting its capacity limits, while regions with intense testing kept a functional healthcare system. The northern regions of Italy faced bottleneck situations in hospitals, which is well reflected by the analysis (Fig. 5c, red color). Especially in Lombardia, where only symptomatic patients were tested, the healthcare system was overwhelmed. In contrast, Veneto performed ~4 times more tests per infected than Lombardia, which reduced the number of infections and hospitalizations.

In summary, the hospitalization and testing data resolved per region suggest the benefit of intense testing strategies to mitigate the load on the healthcare system. The *in silico* experiments add evidence that this relationship is induced by the interruption of infection chains, in particular, by the detection of undetected cases. Thus, testing not only improves knowledge on the infection state but also directly impacts the dynamics of the pandemic.

**Capacity Model and excess dead due to the shortage of hospital beds**. Besides NPIs and promoting social distancing, self-isolation etc., strengthening the healthcare system is also an inevitable part of the government response. According to the data published by the Italian Ministry of Health, Italy had 3.18 beds per 1000 people with an average occupancy of 75-90% before the pandemic[41]. Between March 1st and March 11th, 2020, 9–11% of the infected

people were admitted to ICU. Out of total ~5200 ICU beds (pre-pandemic) in Italy, 2500 were already occupied by March 20th. To cope with this critical situation, each region increased the hospital and ICU facilities (Table 3). Despite the considerable increase in hospital and ICU capacity, the unexpected huge wave of patients and the necessary time to adapt the facilities added to the difficulties of crisis management.

To investigate the impact of the limited healthcare capacities, we developed the *Capacity* and the *MaxCap Model* (Figs. 1 and 2). As of May 22nd, all regions had increased the number of available beds, and the epidemic curves were in a downward phase. Therefore, we investigated a possible impact on the pandemic of hospital overload in the previous months, considering data from February 24th to May 22nd.

In the *Capacity Model*, the pre-pandemic occupancy of hospital beds and ICUs was considered 85% and 50%[41], and the baseline number of available hospital and ICU beds was set to 15% and 50% of the total capacity (Table 3), respectively. The capacity that was increased during the crises was described by a region-specific linear function with a daily increment so that the capacity reached its target value at the date indicated in Table 3. We assumed that the unavailability of hospital or ICU beds leads to faster and more frequent death (see Methods). Parameters were determined using the same protocol as for the *Reference Model*. The *Capacity Model* fit is shown in Fig. 6a and Supplementary Fig. 11.

In the *MaxCap Model*, the hospital and ICU capacities were fixed to their maximum from the very beginning. The compartmental flow in both models was kept identical by using the same parameter set as for the *Capacity Model*. We quantified the impact of the limited capacity on COVID-19-associated deaths by subtracting the death numbers in the *MaxCap Model* from those in the *Capacity Model*. This difference represents the number of people that would have benefited from a system with a substantially higher capacity at the beginning of the epidemic. The effect of limited healthcare facilities was dramatic in Lombardia with a ~26% difference in the number of deaths, corresponding to ~4500 people (Fig. 6b).

**A combined strategy**. In reality, many regions ramped up facilities to test the immediate contacts of an identified infection, e.g., Veneto[52] and also strengthen healthcare facilities to accommodate more patients. Previously we observed the benefits of an early TI strategy that reduces the load upon the healthcare system, thereby reducing death numbers (*Testing Model*, Fig. 5a, b). We also observed a reduction in fatal outcomes in the case of a functional healthcare system that is not overwhelmed (*Capacity Model*, Fig. 6a, b). Given these observations, we sought to investigate the combined effects of an improved healthcare facility with intense testing by considering the following four scenarios:

1. linear increase of hospital capacity combined with early TI;
2. linear increase of hospital capacity combined with late TI;
3. maximum hospital capacity from the beginning combined with early TI;
4. maximum hospital capacity from the beginning combined with late TI.

Thereby, *early/late* was assumed one week before/after the lockdown.

To simulate scenarios 1 and 2, we transferred the *Capacity Model* parameters into the *TestCap Model* (Fig. 2) to keep the compartmental flow intact. To simulate scenarios 3 and 4, we evaluated the *TestCap Model* assuming the hospital and ICU capacity at their maximum levels from the beginning. In all

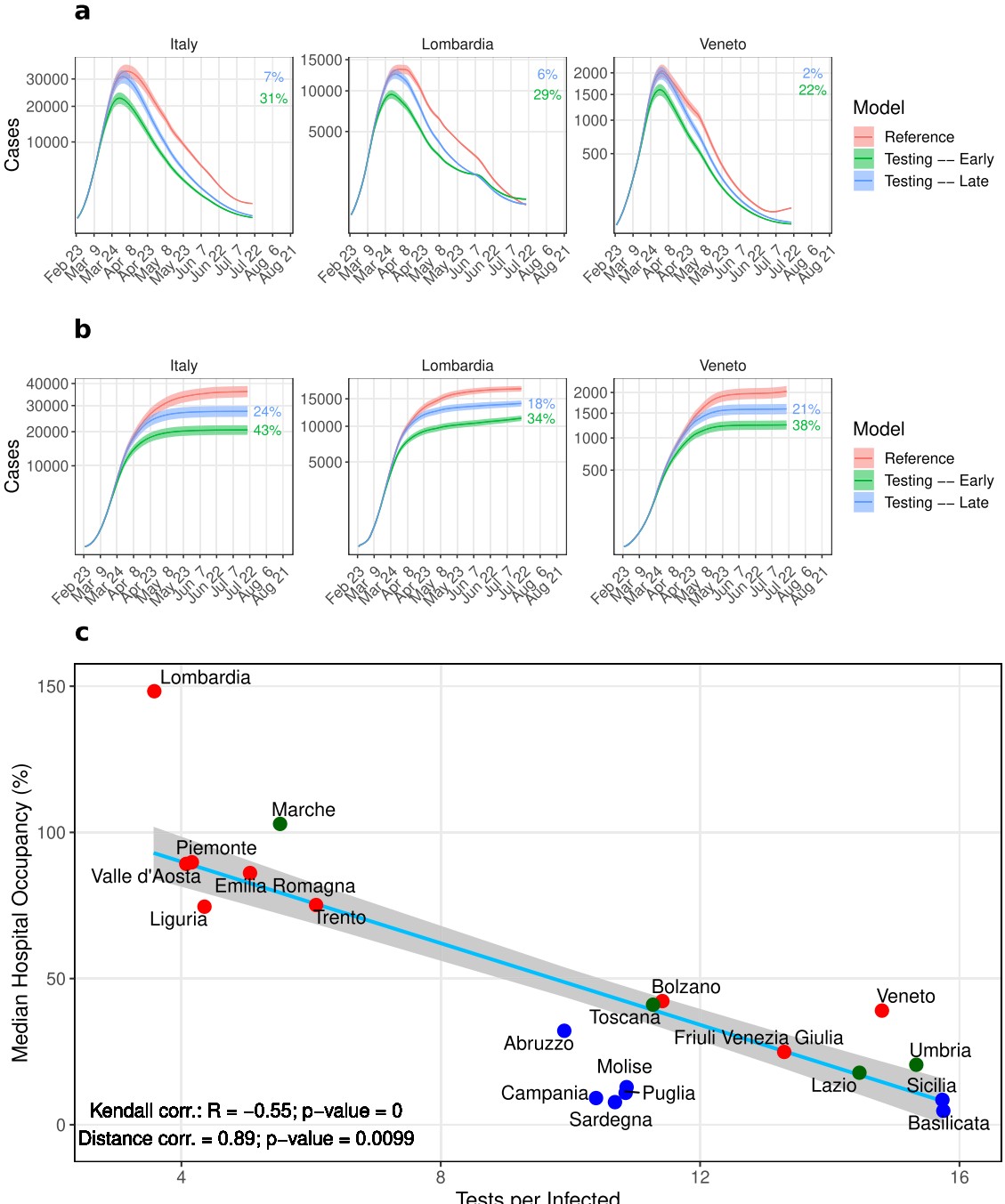

**Fig. 5 Impact of testing and isolation on hospitalization and death. a, b** Simulation results of the *Testing Model* in two scenarios: undetected cases are decreased from $\bar{\mu} = 90\%$ to 60% starting one week before the lockdown (green) and one week post lockdown (blue). **a** Sum of hospitalized including ICU patients. **b** Total deaths. The percentages provided in panels A and B quantify the reduction in peak with respect to the fitted *Reference Model* (red line). Statistics were performed by fitting the data with 100 perturbed parameter sets (see Methods). Continuous line and shaded region represent the mean and standard deviation, respectively. **c** Kendall and Distance correlations have been performed between the number of tests per infected and the median hospital occupancy, defined as the median of daily hospitalized over the pre-pandemic hospital plus Intensive Care Units (ICU) beds. Light blue line: linear regression fit over 20 regions; gray shaded area: standard error; red dots: northern regions; blue dots: southern regions; green dots: central regions. *R*: correlation coefficient; *p*: significance.

scenarios, improved TI was simulated as a step-wise reduction of undetected cases by 2% per day ($\mu'(t)$, *Testing Model* in Methods), which is equivalent to ~10-fold increased detection.

Figure 7 reveals several interesting facets. The importance of testing in a realistic situation where the step-wise extension of the healthcare facility has been installed is depicted in Fig. 7a. It reveals that an early TI strategy brings down hospitalizations

close to the pre-pandemic hospital capacity (horizontal black line). An exception is Lombardia, where adopting an early TI strategy led to ~10% reduction of the hospitalization peak, while the late TI strategy did not decrease the peak size and led to a situation similar to what happened in reality. Though the peak size remained unchanged in Lombardia in the late TI scenario, simulations showed a reduction in the number of deaths of ~17%

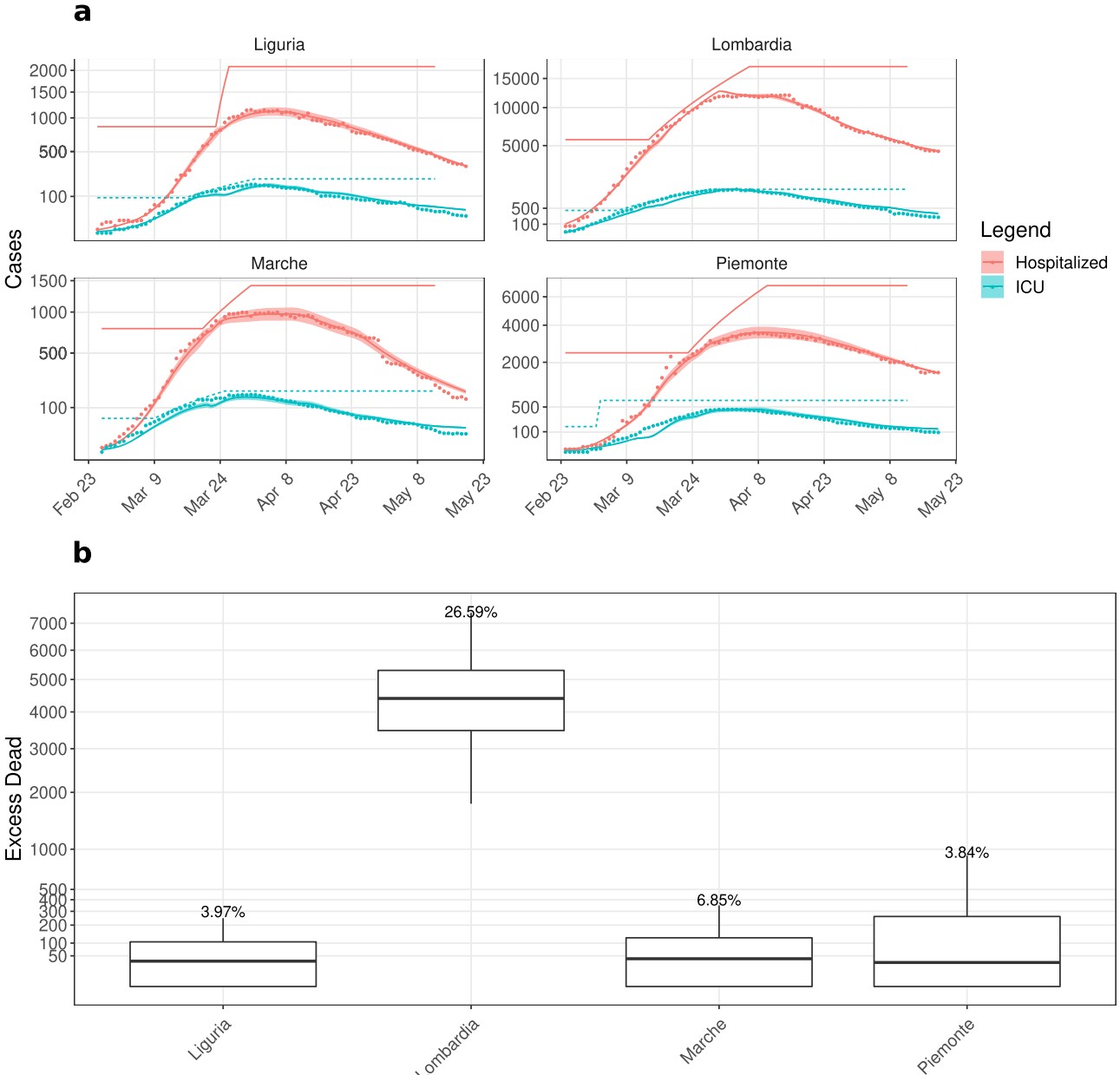

**Fig. 6 Life costs of the limited healthcare system. a** Sample *Capacity Model* fit of hospitalized (red), ICU (iris-blue) data for the most affected regions. Parameters were fitted as in the *Reference Model* by considering the first 3 months (February 24th–May 22nd, 2020) data. Continuous lines: simulation results; round dots: data; Baseline capacity and linear increase in hospital (red) and Intensive Care Units (ICU) (iris-blue) capacity (Table 3) are represented as lines with the respective color. **b** Box plot of the difference in death numbers between the results of the *Capacity* and the *MaxCap Model*. Statistics performed over 100 perturbed parameter sets (see Methods) and the box plot shows the median, 25- and 75-percentiles as well as the minimum and the maximum values. We analysed the variation in the excess dead numbers depending on different values of $\alpha$ for the most affected region, Lombardia (Supplementary Fig. 13).

(Fig. 7a), reaching ~40% (Fig. 7b) when combined with an increased hospital capacity. However, adopting an early TI strategy is effective in reducing the death toll across all regions, ranging from ~34% to ~52%.

Figure 7 B represents the importance of a TI strategy in the settings of a functional healthcare system. Previously, we estimated a 26% reduction in death numbers in Lombardia by strengthening the hospital infrastructure (Fig. 6b). Improved hospital capacity with early TI further reduced death numbers by a substantial amount, which ranged from ~35% to 52% across different regions, ~52% in Lombardia (Fig. 7b). Early TI with ~5-fold more testing would have reduced the death toll up to ~33%

in Lombardia (Supplementary Fig. 12). In the late TI scenario, it would have decreased ~40% in Lombardia (Fig. 7b), and considering only ~5 times more testing it would have decreased deaths by ~28% (Supplementary Fig. 12).

In summary, this *in silico* experiment emphasizes the importance of an early TI strategy that could partially compensate for limited healthcare facilities during the early period (March to May 2020) of the pandemic. However, such an early TI strategy would not be sufficient to contain the hospitalizations within the pre-pandemic hospital limit (horizontal black line, Fig. 7). Therefore extending the hospital infrastructure was mandatory to prevent an overwhelmed healthcare system.

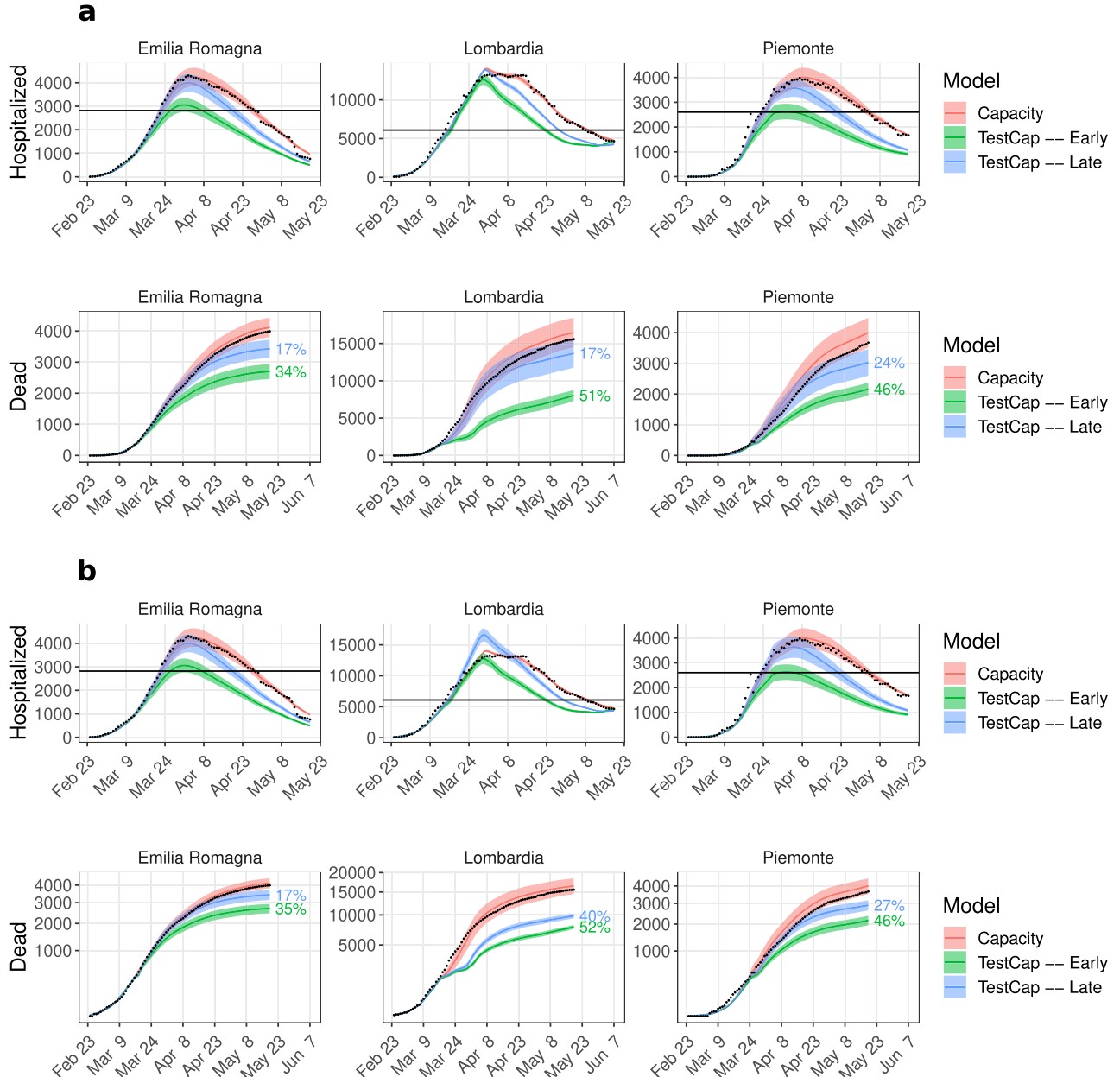

**Fig. 7 A strategy of early TI and extending hospital capacity combined.** Simulation results of the *TestCap Model*. **a** Linear increase of hospital and ICU capacities (scenarios 1 and 2) and **b** maximum capacities from the beginning (scenarios 3 and 4) with early (green) or late testing. Statistics were performed by fitting the data with 100 perturbed parameter sets (see Methods). The line and the shaded region represent the mean and standard deviation, respectively. The hospitalized population is the sum of hospital and ICU patients. The percentage indicates the mean reduction in hospitalized peak and death number with respect to the mean of the *Capacity Model*. Black dots: data; black horizontal line: capacity (hospital + Intensive Care Unit) before the pandemic.

## Discussion

Our retrospective study revealed many lessons that can be learned from the COVID-19 situation in Italy. Though containment measures are necessary to reduce the exponential growth of the pandemic and flatten the infection curve as early as possible, prolonged lockdown (mass quarantine) is not a sustainable solution to the pandemic given its socio-economic burden[53]. The power of lockdown lies in restricting the contacts and reducing the spread mainly from the asymptomatic and pre-symptomatic carriers. This particularly applies to COVID-19 because a substantial part of secondary infections occurs prior to disease onset[54]. Several studies indicate that the silent transmission from pre- and asymptomatic patients was responsible for the majority

of new infections[40,55,56]. Although a recent study has found that there is a lower risk of transmission from asymptomatic people[57,58], asymptomatic cases still present a public-health risk, as they are usually unaware of their infection and do not take any increased self-isolation measures. In Fig. 3 we showed that testing not only reduces the dark figure but also interrupts subsequent transmission chains from undetected cases and thereby directly influences the infection dynamics. This emphasizes the importance of a massive testing strategy to control the pandemic.

With the help of a mathematical analysis of the different regions of Italy, we have provided evidence for a general relation between intense testing and reduced burden on the healthcare system. To interrupt virus transmission chains, the Veneto

Region developed a comprehensive public health strategy focusing on case finding, contact tracing and quarantining close and occasional case contacts[59]. Moreover, testing was extended to both symptomatic and non-symptomatic case contacts. Toscana followed a similar strategy as Veneto, whereas Lombardia tested only the symptomatic cases. Lombardia has twice the population of Veneto (10M vs 5M). Tests performed per capita in Veneto were almost twice as high as in Lombardia[60]. The epidemiological outcomes of the testing strategy adopted by Veneto and Lombardia were different in terms of incidence number, evolution of the pandemic and bottleneck situation of the healthcare system. Veneto and Toscana flattened the infection curve one month earlier than Lombardia and Piemonte. In Fig. 3, Lombardia, Emilia, and Piemonte are in the low test, high infection regime, whereas Veneto is in the high test and low infection regime. Therefore, the pandemic situation in different Italian regions must be addressed in light of their testing strategy. Different testing strategies and their implications for the reproduction number have been studied[55,61]. Several other studies emphasize the combination of contact tracing with testing and its implication on the incidence number[50,62]. Our results suggest that the bottleneck in the healthcare system of northern Italian regions was a consequence of their TTI strategy. In particular, testing of symptomatic individuals alone appears inefficient.

In the middle of the crisis, many healthcare workers were infected with COVID-19 while treating COVID-19 patients, and the voluntary participation of interns and retired personnel was required. Moreover, delays in the testing of healthcare personnel led to the spread of infection through healthcare workers. The shortage of healthcare workers together with limited hospital beds led to a bottleneck situation in the healthcare system. A study revealed that a weekly screening of the healthcare workers and other high-risk groups irrespective of their symptoms would reduce transmission by 23%[61]. We quantified the impact of the overwhelmed healthcare system on the death toll and studied how such a bottleneck situation could be avoided. We showed that a substantial portion of the death toll, ~35% in Lombardia, could be prevented by testing, and this could have mitigated the shortage of healthcare facilities at the early stage of the pandemic, though it would not have contained the hospitalization within its pre-pandemic limit in all cases. Hence, given the capacities, the bottleneck of healthcare facilities could not be completely avoided by adopting massive testing alone.

A ~10-fold increased testing demands huge testing facilities and is not an economic strategy. Moreover, successful detection depends on the subjects getting tested and their previous travel and contact history. Alternatively, tracking and targeted testing with the home quarantine of possible cases substantially reduce infection transmission and could be adopted instead of rapid mass testing due to the higher chance of successful detection. However, manual contact tracing is infeasible at higher incidences, and contact tracing apps require ethical clearance in accessing the location data, transparency and protection of personal data[55]. In this context, it has to be emphasized that, in the framework of our mathematical model, the impact of tests on the spreading of infections is based on the isolation of positively tested individuals regardless of their symptoms, in line with previous observations[50,56]. This implies that contact tracing and quarantine without testing would have a similar effect and might be an efficient strategy when sufficient test capacities are not available. Thus, an effective contact tracing and quarantine mechanism, monitored through modern technologies, together with improved healthcare facilities could reduce mortality in the possible future waves or other pandemics.

Our modelling study has several limitations. During the first wave in Italy, there was a high degree of uncertainty regarding the fraction of pre-symptomatic and asymptomatic cases and their associated transmission. These fractions were also subject to the regional testing strategy, and their dynamic nature is observed in the weekly reports published by ISS. In our analysis, we set the asymptomatic fraction, $\alpha$, to the national average, 0.4. In our simulation, we considered a fixed incubation period. We performed a sensitivity analysis of $R_3$ upon the $\mathcal{R}_t$ within the range provided in Table 1, and we found that $\mathcal{R}_t$ is less sensitive towards the variations of $R_3$.

As mentioned in the *Parameterization* section, we determined the range of parameter values that constrained the fit of the physiological parameters by the information available in the literature on the virus characteristics. Some of the physiological parameters in the model may be internally linked. For example, hospitalization and ICU cases increase with infection cases. Here, we assumed that the nature of the virus did not change considerably during the investigation period, and, therefore, we kept the physiological parameters, and hence the internal relations, constant throughout the investigation period. The behavioral adjustments (through NPIs and awareness) caused the breakdown of such relationships and shaped the pandemic. As the behavioral parameters reflect the public behavior which evolved with time, they depend upon several factors and are region-specific. Therefore, it is difficult to infer a proper distribution of such parameters. Nevertheless, as a more precise parameter distribution would become available, sampling from a different distribution might improve the quality of the model.

For the present analysis, we did not use an age-stratified version of the *Reference Model* due to the lack of knowledge of age-stratified model parameters and incomplete age-stratified data during the first wave. However, the age-dependence is phenomenologically included in the model by using a time-dependent hospitalization rate, which reflects the demography of the infected people and other contingent factors that might alter the outcome of the pandemic. We also did not include co-morbidities or pre-existing medical conditions of a sub-population and did not explicitly consider the potential changes in viral transmissibility due to environmental factors, such as temperature and humidity. Further, cross-regional movements and the potential for imported or exported infections, which might hamper the testing strategy based on contact tracing, were not considered. Lastly, a delay in testing results might hamper outcomes in a contact tracing-based testing strategy[63]. In our simulation, we did not explicitly consider delays in isolation and their impact on the daily cases. Instead of addressing the above limitations explicitly, we perturbed our behavioral parameter sets up to 10% of its base value to ensure the robustness of our results in a plausible range of parameter values.

During the first wave of COVID-19, hospital and ICU beds got overwhelmed in Italy. There are potentially many factors, such as infections from undetected index cases, early vs late testing strategies, limited healthcare facilities, etc., that might have aggravated the COVID-19 situation in Italy. In this paper, we developed a COVID-19 specific infection epidemic model to address the bottleneck situation of the healthcare system that most of the northern regions of Italy, particularly Lombardia have faced during the first wave of the pandemic. As the testing was minimal at the beginning of the pandemic, a large portion of cases remained undetected, which was a major driver of new infections. We first estimated the dark figure for different regions of Italy through a Bayesian Markov Chain Monte Carlo (MCMC) framework. With an adaptive methodology, we estimated the model parameters by fitting the active cases, hospitalized, ICU, and death data published by the Civil Protection Department, Italy. We showed that testing directly influences the infection dynamics by interrupting transmission from undetected cases.

We showed that intense testing is associated with a reduced burden on the healthcare facility and, in reality, regional hospitals were less overwhelmed with more testing. By considering regional pre- and post-pandemic hospital and ICU beds we quantify the impact of the overwhelmed healthcare system on the death amount. This impact was highest in Lombardia, which affected ~4000 people. Implementing an early TTI strategy in Lombardia would have decreased overall hospital occupancy, which would have reduced the death toll by ~45%. However, such a strategy would not have kept the hospitalization amount within the pre-pandemic hospital limit. In this context, it is important to keep in mind that the effectiveness of such a strategy lies in the isolation of positively tested individuals regardless of their symptoms. Therefore contact tracing and quarantine without testing would have a similar effect and might be an efficient strategy when sufficient test capacities are not available.

## Data availability

The dataset analysed during the current study are available in the github repository[29] (https://github.com/pcm-dpc/COVID-19). We considered data till June 23rd, 2020. All data used in this study are included in the GitHub repository, https://github.com/arnabbandyopadhyay/COVID-19-in-Italy inside Data folder, and have been archived on Zenodo[46]. Source data for generating the figures are available inside the 'PARALLEL/Reference_Asymptomatic' and 'PARALLEL/Capacity' folders. Additional instructions to generate figures are given in the readme file in those folders.

## Code availability

The code used for the analysis is available in the GitHub repository: https://github.com/arnabbandyopadhyay/COVID-19-in-Italy, and has been archived on Zenodo[46].

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

## Acknowledgements

Our sincere thanks to Dr. Luciano D'Alfonso, Senatore della Repubblica Italiana, Dr. Angelo Borrelli, capo del Dipartimento della Protezione Civile Italiana e commissario per l'emergenza Covid-19, Dr.ssa Eliana Mazzaro, Dip. della Protezione Civile Italiana, Dott. Stefano Marro, Prevenzione Sanitaria presso Ministero della Salute, for responding to our requests in time and for sharing data of hospital and ICU capacities. We thank Rebecca Ludwig for thoughtful discussion and comments on the manuscript. This project has received funding from the European Union's Horizon 2020 research and innovation programme under grant agreement No 101003480 and the Initiative and Networking Fund of the Helmholtz Association. This project was supported by German Federal Ministry of Education and Research for the project CoViDec (FKZ: 01KI20102). M.S. and A.B. were supported by the COSMIC Marie Sklodowska-Curie grant (765158). TM and SK were supported by the Helmholtz-Gemeinschaft, Zukunftsthema "Immunology and Inflammation" (ZT-0027); from 2021: (ZT-I-0010). The funding bodies had no role in the design of the study, collection, analysis, and interpretation of the results, or writing the manuscript.

## Author contributions

A.B. and M.S. acquired and pre-processed the data, coded the simulation, automated the analysis of the data, analyzed simulation results and led the work. T.M. and M.M.H. developed the reference SECIRD model. A.B., M.S., and T.M. designed the capacity model and protocols for parameter estimation of SECIRD models. A.B. and M.S. developed other versions of the SECIRD model. T.M. derived the literature based parameter sets and analytical expressions of the reproduction numbers of the models. T.M. and S.K. provided insights through discussion. A.B. wrote the paper. A.B., M.M.H., M.S., and S.B. revised the paper. M.M.H. supervised the study.

## Funding

## Competing interests

The authors declare no competing interests.
