## [Peer Review File · Communications Medicine]

Reviewers' comments:

Reviewer #1 (Remarks to the Author):

Dear Editor,

I have read the manuscript titled "Cluster isolation and testing to prevent overloaded health care facilities and to reduce death rates in the SARS-CV-2 pandemic in Italy", by A Bandyopadhyay et al. The subject is undoubtedly important and their methodology novel. They make use of the rich data available for Italy to try to understand whether the approaches used in different Italian regions had an impact on the disease burden, providing an important insight into the matter through direct comparison between geographic areas subjected to the same governmental action, but with different testing approaches. The main outcome points to the importance of Test and tracing in the control of COVID-19 and the reduction of the disease burden. There has been indeed much discussion on the capability of test and trace to be able to put the COVID-19 pandemic under control, and this looks as a relevant contribution. The manuscript is generally well written, apart from some issues that I detail below. My main concern of the paper involves the fitting methodology, and is probably due (I hope) to a bit of confusion in the manuscript. My comments follow:

Major comments:

1) At Page 8 and Page 11. I found a bit confusing how the model was parametrized. At page 11 the authors explain that they fitted the Reference model and then they used it to estimate R_t . How did they do this? Is this the same as mentioned in Section "Parametrization" where they explain that they fit R_2, \dots, R_9 from the first 2 weeks of exponential growth? In this cases the repetition is confusing. They might add "as explained in the Methods/Parametrization section". And in the latter, what happens to the other parameters? Aren't the authors fitting those also? Or are they fixing some of the values?

In general, I am concerned about the capability to fit all the parameters from two weeks of early growth data. For instance, looking at cases, in early growth you can estimate the growth rate. The growth rate is a function of multiple parameters (in a SEIR model, transmission rate, recovery rate, latent rate), thus I cannot see how this relationship can be disentangled without extra information or going at least beyond the exponential growth phase. Now, the authors are using 4 datasets (cases, hospitalizations, intensive care, deaths), but these sets are obviously related and the model uses several more parameters. They show in the Supplementary Material that they get identifiability, thus I have the feeling that I am missing something in their methodology. Maybe they should provide a bit more of detail and most of all clarity about what exactly they are doing in order to dispel this concern. The Parametrization section should contain a more accurate account of how the fitting are performed, the initial condition used, and the way the different models mentioned are used. Some of this information is scattered around the paper which makes the reading confusing.

2) Authors fix the asymptomatic fraction α to 40%. The choice seems connected to the study of Vo Euganeo which, however, had large confidence intervals. Additionally the asymptomatic fraction has been estimated in various studies around the world providing a range of outcomes and some studies have suggested that it is actually age dependent (see for instance, but not exclusively, Davies et al, Nature Medicine 26, 1205 [2020]). Thus a fraction 0.4 is actually a mean value with a large

error bar, which might be representative of a specific population age profile non necessarily typical of other parts of Italy. Since the choice of alpha determines the ratio between detected and undetected cases, I expect that it will also influence many of the numbers that come out of this work, including the number of deaths that could have been averted. In view of this, why a sensitivity analysis of the result on alpha was not taken into consideration? The authors should provide some insight on how consistent their results are relative to alpha.

Other comments:

3) They should specify when the fitting starts: from the plot it seems they are using the 23rd of February as day zero of their simulations. How is the initial condition constructed? Are they also fitting it? In particular, how did they specify the initial number of undetected cases that were circulating at 23rd of February (if that is the starting date)?

4) Page 4: In eq. 1 the authors used various parameters without saying anything about them. Parameters are detailed in Table 1, thus there should be, as a minimum, a reference to the table in the context of the equation for R_0 .

5) Page 5: The parameter μ is not defined in Table 1

6) Page 6: The authors explain that they introduce a time dependent μ' that starts from μ and is decreased up to 60%. There is no explanation why. Only at page 17 the origin of this, viewed as hypothetical scenario, is explained. At least this is what I understood. The authors should anticipate here the reason why the choices on μ' are introduced, and their use as an hypothetical scenario.

7) In the Capacity model, the two expressions f_{Hlim} and f_{Ulim} are unclear. Looking at page 7, it looks as if they are taking the 10th power of three terms of each equation, which does not really make sense. Looking at the supplementary material it seems that the "10" is not a "ten" but a one-zero choice. This is a confusing notation. Instead of elevating to power "10", elevate to some parameter, say γ , and then specify $\gamma=1$ if... and $\gamma=0$ if... This should be done in both the main text, figure 1 and the Supplementary Material. Also, the equation in the Supplementary Material and those in the main text and figure 1 differ. In any case, why using a 1-0 index? It would be clearer if you used a two-cases expression:

$$f_{Ulim} = \{ 0.5 \text{ if } U_{lim} > U_D + U_R; \exp() / (1 + \exp()) \text{ otherwise } \}$$

and similarly for f_{Hlim} . (I am assuming that f_{Ulim} is set to 0.5 if capacity is exceeded). By the way, at first sight it seems to me that the daily chance of acceptance per individual after capacity is exceeded in Intensive Care should be roughly close to the number of daily deaths + daily recoveries in Intensive care divided by the number of daily new cases: as one place becomes available, it can be occupied by a new patient; similarly for hospitalizations. This would avoid to assign non-existent beds.

8) Page 11 The Asymptomatic model should be listed/added to the discussion over the models in Methods section, as it instead appear here out of the blue.

9) Page 15. second half of page. Is the discussion referring to Fig 5A? If it is then the author should

reference the Figure.

9b) Page 17, Reference to Fig 5B should be Fig 5A? Otherwise Fig 5A is unreferenced in the text. Figure 5A caption: it states that parameters were fitted over the first three months. Why did the authors choose here a different approach with respect to the approach explained in the methods (using 2 weeks of exponential growth) ? They should report this in the main text: all fitting methods and choices should be found under a common section of the paper. Again, I might have missed it in the main text, but otherwise the methodology used should be explained in the Methods section and not scattered around.

Reviewer #2 (Remarks to the Author):

A. Brief summary of the manuscript

The authors consider the first wave of the SARS-CoV-2 pandemic in Italy, in particular the Northern Italian regions. They aim to investigate the impact of test-trace-isolate (TTI) interventions on hospital capacity and death rates using a compartmental model. By combining model with data from the Italian situation, they provide estimates to how changes in parameter values related to TTI scenarios impact hospital occupancy.

B. Overall impression of the work

The authors try to address a lot of different things in their manuscript, including the fraction underreported cases, hospital capacity, early testing, contact tracing. As a consequence it is hard to understand what the main thread of the manuscript is. Furthermore, key assumptions and definitions are not clearly worked out in the main text, e.g. what is the definition of the infection fatality rate (IFR)? Especially in the case of TTI, which seems to be the main focus of the manuscript, it is never explained what is exactly understood by that and how it is implemented in the model, i.e. which assumptions are associated to the TTI intervention. The way the methods and results sections are written make it hard to disentangle which parts in the manuscripts are model assumptions, data analysis, results, or conclusions drawn from other studies. This makes it hard for the reader to verify the validity of the study. Finally, the authors derive a lot of estimates in their study, e.g. the fraction of underreporting, infection rate, IFR, but I miss how they compare to estimates in literature, and this makes it hard to put their study in the right perspective.

C. Specific comments, with recommendations for addressing each comment

1. It is not clear to me how/what clusters are involved in the manuscript. Although the term is prominent in the title of the manuscript, no reference is made in the abstract or main text to clusters. I would recommend revising the title as it can be somewhat misleading.
2. One of the main research questions of this manuscript is how TTI intervention impact hospital occupancy and death rates. Although the authors speak of more frequent testing, earlier testing, and testing a broader population, it is not easy to distill from the text how testing strategies translate into the model. In fact, testing alone wouldn't alter an epidemic, rather the effects testing has, e.g. in isolation of positively tested individuals.
3. Data and code availability: are data and code available? I expect them to be for submitting to the journal, but can't find it back in the data/code statement in the manuscript. At the moment it would not be possible to reproduce the results in the manuscript.
4. The authors seem to be concerned only with the absolute time in the epidemic when comparing regions, e.g. May 22nd 2020 is considered as a cut-off date, implicitly assuming that all regions in Italy were at the same point in the epidemic outbreak at all times. Please make this assumption explicit in the text and give a justification for making such an assumption.

5. The compartmental model used by the authors seem to assume a fixed deterministic time in each compartment, e.g. by way of descriptions of parameters in table 1: $1/R_3$ is the pre-symptomatic infectious period. However the assumption on fixed times in compartments is not made explicit. Moreover, it would be helpful if the authors pay attention to this assumption in the limitations section.

6. As I see it, the model used by the authors is in many ways a classical epidemic compartmental model, such as the SIR, SEIR compartmental models that the authors compare to, albeit with more compartments. I believe the point the authors try to make in the comparison with SEIR is that they formulated a model that is tailored to the epidemic and questions, i.e. regarding hospital capacity, and not to discard an entire suite of epidemic models. I would recommend the authors to rephrase the text regarding compartmental models.

7. It is unclear which parameters are time-varying and which are not, and what the justification for each of them are. The authors state several times that in the course of the epidemic behavioural changes and NPIs impact certain parameters and therefore they should be time varying. At the same time the infectious period remains constant throughout, shouldn't that by the same reasoning also be time varying? Please make clear why certain choices are made, and include them in the parameter overview tables for ease of the reader.

8. Please pay attention to when parameters are introduced in the text, it is hard for the reader to piece together all the information. For example, the R_x parameters are presented in table 1 but they appear for the first time in formula (1) without any reference in the text.

9. The equations presented in the main text do not contain enough information for the reader to verify their derivation. For example, the R_0 and $R(t)$ in the method section seem to play an important part in the analysis, but how does the $R(t)$ on page 7 relate to formula (2)? Or the expressions f_{Hlim} , f_{Ulim} are stated in the section "capacity model" without further explanation.

10. The authors do a sensitivity analysis by perturbing the values from the main analysis for a subset of parameters. It is claimed that in this way confidence intervals for R_t are generated. As confidence interval is largely understood as a statistical concept, to which also a significance level is attached, I believe that the authors do not mean confidence intervals in their case. If this is the case however, it should be shown that their method indeed yields confidence intervals (and if so, with what confidence level?).

Dear reviewers,

Thank you for giving us the opportunity to submit a revised version of our manuscript entitled formerly, "Cluster isolation and testing to prevent overloaded health care facilities and to reduce death rates in the SARS-COV-2 pandemic in Italy" to Communications Medicine. We appreciate the time and effort the reviewers have dedicated to our paper, and we thank all the reviewers for the constructive criticisms and insightful comments to improve the current version of our manuscript. We have been able to incorporate changes to reflect all of the suggestions provided by the reviewers. We have highlighted the changes within the manuscript.

Here is a point-by-point response to the reviewers' comments and concerns.

Reviewer #1 (Remarks to the Author):

Dear Editor,

I have read the manuscript titled "Cluster isolation and testing to prevent overloaded health care facilities and to reduce death rates in the SARS-CV-2 pandemic in Italy", by A Bandyopadhyay et al. The subject is undoubtedly important and their methodology novel. They make use of the rich data available for Italy to try to understand whether the approaches used in different Italian regions had an impact on the disease burden, providing an important insight into the matter through direct comparison between geographic areas subjected to the same governmental action, but with different testing approaches. The main outcome points to the importance of Test and tracing in the control of COVID-19 and the reduction of the disease burden. There has been indeed much discussion on the capability of test and trace to be able to put the COVID-19 pandemic under control, and this looks as a relevant contribution. The manuscript is generally well written, apart from some issues that I detail below.

My main concern of the paper involves the fitting methodology, and is probably due (I hope) to a bit of confusion in the manuscript. My comments follow:

Major comments:

1) At Page 8 and Page 11. I found a bit confusing how the model was parametrized. At page 11 the authors explain that they fitted the Reference model and then they used it to estimate R_t . How did they do this? Is this the same as mentioned in Section "Parametrization" where they explain that they fit R_2, \dots, R_9 from the first 2 weeks of exponential growth? In this cases the repetition is confusing. They might add "as explained in the Methods/Parametrization section". And in the latter, what happens to the other parameters? Aren't the authors fitting those also? Or are they fixing some of the values?

Authors' response: Thank you for pointing this out. We now cleared the repeated description of the fitting procedure and rewrote some parts of the **Methods section > Parameterization paragraph (line n. 118-144)**

In our parameter estimation procedure, we first estimated the physiological parameters (within their ranges reported in Table 1) by fitting the first 2 weeks of the pandemic data. The value obtained for each parameter was fixed throughout the subsequent fitting. In the second step, to estimate the behavioural parameters, we employed a moving time window technique with a window size of 7 days. Each behavioural parameter was fitted in each time window. These parameters were then used to calculate R_t using the formula in eq.2. Thus, given the parameter fit, R_t is a function of those parameters and not subject to an extra estimation procedure. We have now clearly mentioned it in the **Results Section > Influence of undetected cases on R_t** (line n. 351-356).

In general, I am concerned about the capability to fit all the parameters from two weeks of early growth data. For instance, looking at cases, in early growth you can estimate the growth rate. The growth rate is a function of multiple parameters (in a SEIR model, transmission rate, recovery rate, latent rate), thus I cannot see how this relationship can be disentangled without extra information or going at least beyond the exponential growth phase.

Authors' response: Thank you for the comment and the opportunity to clarify this aspect.

We have used the first two weeks of data in cases, hospitalized, ICUs and dead in order to estimate parameters. Some parameters were not estimated or had additional constraints to fulfill, which narrows down the parameter search space. For example, α was set to the national average of 0.4, $\bar{\mu}$ was estimated through the MLE method of a Bayesian framework (in the supplementary) and was fixed, $\frac{1}{R2} + \frac{1}{R3} = 5.2$ days, $\frac{1}{R9} = \frac{1}{R3} + \frac{1}{2R4}$.

We agree with the reviewer that the growth rate is a function of several factors, even though the exact nature of this relationship is unknown. Therefore, disentangling all related parameters would be difficult and speculative without passing extra information. In Italy, after two weeks, lockdown and other measures were implemented and using data beyond these two weeks would further complicate the situation.

We included the explanation in the **Methods Section > Parametrization** (line n. 127-138) and in the **Discussion Section > Limitations** (line n.593-602).

In the initial phase (we considered the first two weeks), though, when the public was little aware of the disease spreading, and no NPIs were in place, the growth rate mainly depended on the viral properties (along, of course, with other properties like population density and contact rates).

On the other hand, awareness and NPIs indeed break the relationship between growth rate and viral properties through contact reduction and impact the behavioural parameters, but with a delay that depends upon the epidemic state and implementation time.

Therefore, we used this 'socially unaware' period (first two weeks) to estimate the physiological parameters (R_x , $x = 2, \dots, 9$) which reflect viral properties. These parameters were then fixed throughout the simulation, as we assumed the nature of the virus would not change significantly during the investigation period. This strategy intrinsically/inherently assumes that the internal relations remained constant throughout the course of the

investigation, and the behavioural adjustments (through NPIs and awareness) caused the breakdown of such relationships and shaped the pandemic.

To capture the effect of the NPIs and public awareness, we used the 7-days window fitting strategy and estimated the behavioural parameters ($\rho, \vartheta, \delta, R_1, R_{10}$) in each time window.

Now, the authors are using 4 datasets (cases, hospitalizations, intensive care, deaths), but these sets are obviously related and the model uses several more parameters. They show in the Supplementary Material that they get identifiability, thus I have the feeling that I am missing something in their methodology. Maybe they should provide a bit more of detail and most of all clarity about what exactly they are doing in order to dispel this concern. The Parametrization section should contain a more accurate account of how the fitting are performed, the initial condition used, and the way the different models mentioned are used. Some of this information is scattered around the paper which makes the reading confusing.

Authors' response: Even though we estimated many parameters, some were fixed or have additional constraint (e.g. asymptomatic factor, $\alpha = 0.4$, incubation period, $\frac{1}{R_2} + \frac{1}{R_3} = 5.2$ days, $\frac{1}{R_9} = \frac{1}{R_3} + \frac{1}{2R_4}$, undetected fraction, $\bar{\mu}$), which narrows down parameter search space. Along with the fixed parameters, data in cases, hospitalizations, ICUs, and deaths were fitted simultaneously to get the rate or the fraction that is entering into a compartment.

We agree that the four datasets we used to parameterize the model are interlinked. For example, hospitalization is a fraction of total cases (active cases+new cases) and ICUs can also be considered a fraction of hospitalizations. Therefore, such relations could be used to guide the compartmental flow. As we are fitting these datasets to set the compartmental flow, the same relationship can also be inferred from the compartmental model given that it fits the data, which we showed in Fig 2.

We have now added the **Methods Section > Initial condition paragraph (line n. 104-116)** in the Methods Section and rewritten the **Methods Section > Parameterization paragraph (line n. 118-144)**.

2) Authors fix the asymptomatic fraction alpha to 40%. The choice seems connected to the study of Vo Euganeo which, however, had large confidence intervals. Additionally the asymptomatic fraction has been estimated in various studies around the world providing a range of outcomes and some studies have suggested that it is actually age dependent (see for instance, but not exclusively, Davies et al, Nature Medicine 26, 1205 [2020]). Thus a fraction 0.4 is actually a mean value with a large error bar, which might be representative of a specific population age profile non necessarily typical of other parts of Italy. Since the choice of alpha determines the ratio between detected and undetected cases, I expect that it will also influence many of the numbers that come out of this work, including the number of deaths that could have been averted. In view of this, why a sensitivity analysis of the result on alpha was not taken into consideration? The authors should provide some insight on how consistent their results are relative to alpha.

Authors' response: We agree that the national average of the asymptomatic fraction α might have a large error bar and might vary strikingly among the regions as the regional settings might differ in terms of testing policy, the pandemic phase, etcetera. Importantly, regional estimation of the asymptomatic fraction is also mathematically challenging given the limited resources. Computationally, we can estimate the variation in results for a choice of α by doing a sensitivity analysis, and we found that R_t is not very sensitive to the variation in α (supplementary Fig. S9). We now analysed the variation on the excess dead numbers (Fig.5B) depending on different values of α for the most affected region, Lombardia. We included this into the supplementary material (Supplementary Fig S13).

***Other comments*:**

3) They should specify when the fitting starts: from the plot it seems they are using the 23rd of February as day zero of their simulations. How is the initial condition constructed? Are they also fitting it? In particular, how did they specify the initial number of undetected cases that were circulating at 23rd of February (if that is the starting date)?

Authors' response: Thank you for pointing it out: we have now clarified these details in the **Methods Section > Initial Conditions (line n. 104-116)**.

The Italian data started on February 24th, 2020, and the simulation started 5.2 days earlier (incubation time). We assumed that the total cases reported on February 24th plus the undocumented cases (estimated as detailed in the supplementary) were exposed 5.2 days earlier. We assumed all other compartments to be 0, except for the Susceptible which were set to the population size.

E.g., The first reported case number in Italy was 100, and we estimated 90% undetected cases. Therefore, 100 represents 10% of reported cases, and 900 are undetected cases. The simulation begins from -5.2 days, with susceptible as the Italian population and 1000 exposed (100 detected + 900 undetected).

4) Page 4: In eq. 1 the authors used various parameters without saying anything about them. Parameters are detailed in Table 1, thus there should be, as a minimum, a reference to the table in the context of the equation for R_0 .

Authors' response: Thank you for pointing this out. We have now referenced Table 1 and described the parameters.

5) Page 5: The parameter mu is not defined in Table 1

Authors' response: Thank you for pointing this out. We have now added mu in Table 1.

6) Page 6: The authors explain that they introduce a time dependent mu' that starts from mu and is decreased up to 60%. There is no explanation why. Only at page 17 the origin of this, viewed as hypothetical scenario, is explained. At least this is what I understood. The authors should anticipate here the reason why the choices on mu' are introduced, and their use as an hypothetical scenario.

Authors' response: Thanks to the reviewer for this comment. Mu' is conceptualized in order to capture the effect of the enhanced testing and isolation policy of reducing the dark number and the impact of isolation upon the new positive cases. We have now justified our choices in the **Methods Section > Testing Model (line n. 225-231)**.

7) In the Capacity model, the two expressions f_Hlim and f_Ulim are unclear. Looking at page 7, it looks as if they are taking the 10th power of three terms of each equation, which does not really make sense.

Looking at the supplementary material it seems that the "10" is not a "ten" but a one-zero choice. This is a confusing notation.

Instead of elevating to power "10", elevate to some parameter, say gamma, and then specify gamma=1 if... and gamma=0 if... This should be done in both the main text, figure 1 and the Supplementary Material. Also, the equation in the Supplementary Material and those in the main text and figure 1 differ. In any case, why using a 1-0 index? It would be clearer if you used a two-cases expression:

$$f_Ulim = \{ 0.5 \text{ if } Ulim > U_D+U_R; \exp()/(1+\exp()) \text{ otherwise } \}$$

and similarly for f_Hlim. (I am assuming that f_Ulim is set to 0.5 if capacity is exceeded). By the way, at first sight it seems to me that the daily chance of acceptance per individual after capacity is exceeded in Intensive Care should be roughly close to the number of daily deaths + daily recoveries in Intensive care divided by the number of daily new cases: as one place becomes available, it can be occupied by a new patient; similarly for hospitalizations. This would avoid to assign non-existent beds.

Authors' response: Thank you for this comment. We clarified this point and the purpose of the functions f_Ulim and f_Hlim in the **Methods Section > Capacity Model (line n. 259-265)**.

The purpose of the capacity model is to keep the influx intact in the hospital or ICU as long as the hospital and ICU beds are available (i.e. $(H_R + H_U < \text{Hosp.Beds})$ and $(U_R + U_D < \text{ICUBeds})$). Upon reaching the capacity limit, the influx would be stopped until a vacant bed is available. As the reviewer suggested, this could be achieved by introducing a Heaviside

step function or any other piecewise method, but this type of function introduces discontinuities and makes solving the ODEs computationally demanding and error-prone. The way out is to conceptualize a function that behaves like a step function but is continuous in nature. In the functional form of f_{Hlim} and f_{Ulim} , we used the 10th power to achieve this. f_{Hlim} and f_{Ulim} return 1 as long as there are free hospital or ICU beds and 0 otherwise.

9) Page 15. second half of page. Is the discussion referring to Fig 5A? If it is then the author should reference the Figure.

Authors' response: Thank you for the remark. We now have referenced Fig 6A (which was previously 5A) in the **Results Section > A combined strategy (line n.452,469)**.

9b) Page 17, Reference to Fig 5B should be Fig 5A? Otherwise Fig 5A is unreferenced in the text.

Figure 5A caption: it states that parameters were fitted over the first three months. Why did the authors choose here a different approach with respect to the approach explained in the methods (using 2 weeks of exponential growth) ? They should report this in the main text: all fitting methods and choices should be found under a common section of the paper. Again, I might have missed it in the main text, but otherwise the methodology used should be explained in the Methods section and not scattered around.

Authors' response: Thank you for pointing this out. Indeed, the reference to Fig.6A (which was previously 5A) was missing and we have now rectified it.

For estimating the *Capacity Model* parameters, we followed the same procedure used for the *Reference Model* by considering the first two weeks of data, but the simulation ends on May 22nd.

The aim of the Capacity Model was to assess whether a timely increase in facilities could reduce the number of deaths.

By May 22nd, all hospitals had increased the number of beds, and the epidemic curves were in a downward phase. Therefore, the excess of mortality due to hospital overloading was sought in the initial phase of the pandemic.

We have now adapted the text to make this clearer: **Results Section > Capacity Model and excess dead due to the shortage of hospital beds (line n.439-443)**.

Reviewer #2 (Remarks to the Author):

A. Brief summary of the manuscript

The authors consider the first wave of the SARS-CoV-2 pandemic in Italy, in particular the Northern Italian regions. They aim to investigate the impact of test-trace-isolate (TTI) interventions on hospital capacity and death rates using a compartmental model. By combining model with data from the Italian situation, they provide estimates to how changes in parameter values related to TTI scenarios impact hospital occupancy.

B. Overall impression of the work

The authors try to address a lot of different things in their manuscript, including the fraction underreported cases, hospital capacity, early testing, contact tracing. As a consequence it is hard to understand what the main thread of the manuscript is.

Authors' response: We agree with the reviewer's comment that we addressed many interrelated aspects. The main thread of the paper is to address the bottleneck situation of the healthcare facilities, and without considering the contributing factors, a thorough investigation would be incomplete. We clarified this concept in the abstract and introduction section. As northern regions of Italy mainly faced the bottleneck situation, regional settings (e.g. the amount of hospital and ICU beds, dark number, testing policy, ...) also need to be considered. On top of that, these factors are intertwined in nature. We attempted to analyse each aspect by considering parameters associated with those contributing factors and by setting up several models with a particular purpose and comparing them with each other. With this in mind, we believe that the problem statement itself demands addressing multiple issues associated with the main thread. In the flowchart (Figure 3) we summarize different models, key assumptions associated with that model and its purpose.

Furthermore, key assumptions and definitions are not clearly worked out in the main text, e.g. what is the definition of the infection fatality rate (IFR)? Especially in the case of TTI, which seems to be the main focus of the manuscript, it is never explained what is exactly understood by that and how it is implemented in the model, i.e. which assumptions are associated to the TTI intervention. The way the methods and results sections are written make it hard to disentangle which parts in the manuscripts are model assumptions, data analysis, results, or conclusions drawn from other studies. This makes it hard for the reader to verify the validity of the study.

Authors' response: Thanks to the reviewer. We have now defined IFR in the **main text (line n.300)**. IFR estimation method and the comparison of our results with the existing literature are available in the **Supplementary > Bayesian estimation of COVID-19 IFR**.

In the *Testing Model*, we introduced a time-dependent fraction μ' of undetected infections which decreased with time. The detected portion would have reduced the risk of infecting other people. With this model, we reproduce the TTI strategy: testing and tracing leads to a

lower undetected fraction ($\mu'(t)$) and to the isolation of the newly detected cases. We have now clarified this in the **Methods > Testing Model (line n.225-231)** and in the **Results > Test, trace, and isolate (TTI) to reduce hospitalizations (line n.394-398)**.

Finally, the authors derive a lot of estimates in their study, e.g. the fraction of underreporting, infection rate, IFR, but I miss how they compare to estimates in literature, and this makes it hard to put their study in the right perspective.

Authors' response: Thanks to the reviewer for this comment. As the estimation of the IFR is adapted from the method of Rinaldi and Paradisi (2020), we decided to detail the estimation method of IFR in the Supplementary Material. We have also compared our estimated IFR and IR with existing literature and our values are close to the Imperial College London report 20 on Italy (**Results > Region-wise infection fatality rate (IFR) (line n. 317-319)**).

C. Specific comments, with recommendations for addressing each comment

1. It is not clear to me how/what clusters are involved in the manuscript. Although the term is prominent in the title of the manuscript, no reference is made in the abstract or main text to clusters. I would recommend revising the title as it can be somewhat misleading.

Authors' response: Thanks to the reviewer for this comment. What we meant by cluster is the network of infections that would arise if the cases were not early detected and isolated. But we agree with the reviewer and we changed the title with the following: *Testing and isolation to prevent overloaded health care facilities and to reduce death rates in the SARS-CoV-2 pandemic in Italy*.

2. One of the main research questions of this manuscript is how TTI intervention impact hospital occupancy and death rates. Although the authors speak of more frequent testing, earlier testing, and testing a broader population, it is not easy to distill from the text how testing strategies translate into the model. In fact, testing alone wouldn't alter an epidemic, rather the effects testing has, e.g. in isolation of positively tested individuals.

Authors' response: Thanks to the reviewer for this question. In the *Testing Model*, we introduced a time-dependent fraction μ' of undetected infections which decreased with time. The detected portion would have reduced the risk of infecting other people. With this model, we reproduce the TTI strategy: testing and tracing leads to a lower undetected fraction ($\mu'(t)$) and to the isolation of the newly detected cases. We have now cleared this in the **Methods > Testing Model (line n.225-231)** and in the **Results > Test, trace, and isolate (TTI) to reduce hospitalizations (line n.394-398)**.

3. Data and code availability: are data and code available? I expect them to be for submitting to the journal, but can't find it back in the data/code statement in the manuscript. At the moment it would not be possible to reproduce the results in the manuscript.

Authors' response: The link to the code was provided in the Methods section: <https://github.com/arnabbandyopadhyay/COVID-19-in-Italy> (line n.287). We have now also added the link in the **Declarations > Availability of data and materials** (line n.662). We now added a step by step guide to generate each figure.

4. The authors seem to be concerned only with the absolute time in the epidemic when comparing regions, e.g. May 22nd 2020 is considered as a cut-off date, implicitly assuming that all regions in Italy were at the same point in the epidemic outbreak at all times. Please make this assumption explicit in the text and give a justification for making such an assumption.

Authors' response: Thanks to the reviewer for the opportunity to clarify this point. The aim of the Capacity Model was to assess whether a timely increase in facilities could have reduced the number of deaths. By the end of May, hospital and ICU capacities had already increased in all regions and the bottleneck point was surpassed. This amelioration of the situation is apparent from the reduction in the number of hospitalized patients. Therefore, a potential effect due to the overload of the hospitals, had to be investigated in the previous months and we fit data up to May 22nd. We did not assume that regions were at the same point in the epidemic outbreak, and we have now adapted the text to make this clearer: **Results Section > Capacity Model and excess dead due to the shortage of hospital beds** (line n.439-443).

5. The compartmental model used by the authors seem to assume a fixed deterministic time in each compartment, e.g. by way of descriptions of parameters in table 1: $1/R_3$ is the pre-symptomatic infectious period. However the assumption on fixed times in compartments is not made explicit. Moreover, it would be helpful if the authors pay attention to this assumption in the limitations section.

Authors' response: We agree with the reviewer here that the fixed deterministic time spent in each compartment in a single realization presents a limitation. However, we randomly sampled behavioral parameters 100 times within the range mentioned in Table 1 and followed the same procedure. This procedure led to the variation in the parameter set and the resulting dynamics. We mentioned this in the **Methods Section > Perturbation and parameter identifiability** (line n.152-165). We performed the statistics over all the realizations, and therefore results are robust in nature over a range of parameters. We mentioned this in the **Discussion > Limitations** (line n.614-618).

6. As I see it, the model used by the authors is in many ways a classical epidemic compartmental model, such as the SIR, SEIR compartmental models that the authors compare to, albeit with more compartments. I believe the point the authors try to make in the comparison with SEIR is that they formulated a model that is tailored to the epidemic and questions, i.e. regarding hospital capacity, and not to discard an entire suite of epidemic models. I would recommend the authors to rephrase the text regarding compartmental models.

Authors' response: Thank you for this comment. We surely did not mean to discard other types of SIR, SEIR model, but to highlight the potential of our model.

We modified the text to avoid this misunderstanding by adding:

'Even though the general compartmental SIR and SEIR type models are useful in inferring epidemic spread and public health interventions, for COVID-19, where the dark numbers, regional testing strategies, hospital beds, etc. plays an influential role in shaping the pandemic, we need to introduce additional compartments to capture the specific phases of the disease' (line n.52-57).

7. It is unclear which parameters are time-varying and which are not, and what the justification for each of them are. The authors state several times that in the course of the epidemic behavioural changes and NPIs impact certain parameters and therefore they should be time varying. At the same time the infectious period remains constant throughout, shouldn't that by the same reasoning also be time varying? Please make clear why certain choices are made, and include them in the parameter overview tables for ease of the reader.

Authors' response: Thanks to the reviewer for pointing this out and giving us the chance to make it clear.

We included in the behavioural parameters (R_1 , ρ , ϑ , δ and R_{10}) those that get influenced by contingent factors. In particular: R_1 will be influenced by the number of close contacts, which, in turn, varies with public awareness and NPIs implementation. ρ , ϑ , δ and R_{10} were left time-dependent to account for the variation of the cases' demography and delays in the pandemic management due to the lack of knowledge and facilities.

On the other hand, physiological parameters primarily depend on the viral properties. Once estimated, these parameters were fixed throughout the simulation, as we assumed the nature of the virus (and hence the parameters associated with it) did not change significantly during the investigation period.

We mentioned in **Table 1** which parameters were fixed and which were time-varying (**column Comments/References**), and we modified the **Methods > Parametrization (line n.127-138)**, to make our reasoning clearer.

To address the specific example given by the reviewer: we considered a fixed incubation period (latent + subclinical period) of 5.2 days. The infectious period (subclinical+clinical) is a clinical parameter that depends upon the age and health conditions of an individual and is difficult to estimate with epidemiological data. Of note, the infectious period does not depend upon the behavior and therefore shouldn't be time-varying as it is conceptualised in our model. However, in our model, the infectious period depends upon the clinical state of an individual, e.g., asymptomatic recovery timescale = $\frac{1}{R_9}$ and mild symptomatic recovery timescale = $\frac{1}{R_4}$, recovery from hospital and ICU are also different.

8. Please pay attention to when parameters are introduced in the text, it is hard for the reader to piece together all the information. For example, the R_x parameters are presented in table 1 but they appear for the first time in formula (1) without any reference in the text.

Authors' response: Thank you for this comment, we now rearranged the *Methods* section.

9. The equations presented in the main text do not contain enough information for the reader to verify their derivation. For example, the R_0 and $R(t)$ in the method section seem to play

an important part in the analysis, but how does the $R(t)$ on page 7 relate to formula (2)? Or the expressions f_{Hlim} , f_{Ulim} are stated in the section “capacity model” without further explanation.

Authors' response: Thank you for pointing this out. The $R(t)$ formula (eq.2) is derived as R_0 (eq.1), but is a function of parameters varying in time, while R_0 corresponds to the one value at the beginning of the pandemic. We now cleared this in the **Methods Section > Basic Reproduction Number (line n. 184-191)**, and we included the derivation in the **Supplementary > R_t calculation**.

We also clarified the purpose of the functions f_{Ulim} and f_{Hlim} in the **Methods Section > Capacity Model (line n.259-265)**.

10. The authors do a sensitivity analysis by perturbing the values from the main analysis for a subset of parameters. It is claimed that in this way confidence intervals for R_t are generated. As confidence interval is largely understood as a statistical concept, to which also a significance level is attached, I believe that the authors do not mean confidence intervals in their case. If this is the case however, it should be shown that their method indeed yields confidence intervals (and if so, with what confidence level?).

Authors' response: Thank you for pointing this out. The term confidence interval is relic from the older versions and we agree with the reviewer here that we do not mean confidence interval in this case. Instead, we calculated standard deviation of the R_t values with the perturbed parameter sets. We have now rectified this mistake in the **Methods Section > Basic reproduction number (line n.187-191)**.

Reviewers' comments:

Reviewer #1 (Remarks to the Author):

Dear Editor,

I have read the response of the authors and verified the manuscript. The authors have answered to all points, and clarified them by adapting the manuscript. This includes one point that is not explicitly mentioned in their reply, but I can see the manuscript was modified to account for it. Overall the manuscript is good, and I am happy to recommend it for publication.

Reviewer #2 (Remarks to the Author):

Thank you the authors for addressing previous comments and making changes to the manuscript. They have clarified some important points of earlier confusion and I believe the manuscript to have improved from the previous version. Upon rereading the new manuscript I do however remain with some comments.

Regarding the manuscript itself.

Major comments

1. After rereading the model description and the author response to an earlier comment I am now confused whether the time spent in each compartment is indeed fixed. Don't the authors model the flow between compartments as a system of ODE? In which case, the time in a compartment would be exponentially distributed. Furthermore, the authors explain they sample from parameter ranges 100 times, but is the sampling done uniformly over the range? I believe that at least for some parameters more information is known in literature about appropriate distributions for the parameters.
2. As I understand, the authors consider a random fraction of undetected cases to become detected through testing and subsequent isolation of cases. If I have understood this correctly, then the usage of the terms contact tracing or TTI are not suitable. TTI or contact tracing are generally understood to involve the explicit tracing of contacts of an index case. I don't see how explicit tracing can be investigated when the model does not keep track on who contacted whom.
3. Page 11 lines 316-317: what do the authors mean by the herd immunity threshold of 70%? How do the authors go from the IFR, which they define as the proportion of deaths among infections, to making a statement about immunity on the population level? Moreover, there should at least be some references to the literature on their quoted herd immunity.
4. I appreciate that the authors try to put their work using a compartmental model in a larger context. However, I believe that they should put more work in putting the context to using compartmental models in the COVID-19 pandemic besides the studies that are cited for Italy specifically. Moreover, I would suggest for the authors to rephrase their text: any research question require a specific model with suitable compartments. Rather they could put the emphasis on their

specific research question, for which they developed the tailor-made model.

Minor comments:

5. Please provide more detail in the references to the ISTAT data: could the authors provide a reference to the urls that they have used and the dates at which they have accessed the pages.

6. What are the consequences of assuming a fixed incubation period? Have the authors performed a sensitivity analysis.

7. Could the authors report on the number of data points in the first two weeks, to which they fit their model?

8. Could the authors comment on the moving time window of seven days? Is this chosen because of the calendar week or data streams?

9. In practice, there are often delays in detection and isolation of cases, which are generally important factors in the effectiveness of these interventions. It would be good for the authors to discuss these in the limitations of their model.

Thank you for the authors for pointing out the link to the github page containing the model code. Regarding the code:

10. I have been unable to find a demo of the code as described in the software statement

11. It would be good to fix the formatting of the README in both PARALLEL/Reference_Asymptomatic and PARALLEL/capacity: save the README as a markdown file and format the text so that it is easier to read

12. What is the expected run time (and on what type of compute infrastructure, e.g. personal laptop), please report as also described in the software statement

13. Could you elaborate on the data files that are used in the code? Where and when (i.e. which files/functions) are the data being used? Furthermore, what are the minimal requirements for the data, say someone would like to run the model on their own data, is that possible under the current implementation, what are the requirements for e.g. column names. These questions should ideally be elaborated upon in a demo or README.

Dear reviewers,

Thank you for giving us the opportunity to submit a revised version of our manuscript entitled, "Testing and isolation to prevent overloaded health care facilities and to reduce death rates in the SARS-COV-2 pandemic in Italy" to Communications Medicine. We appreciate the time and effort the reviewers have dedicated to our paper, and we thank all the reviewers for the constructive criticisms and insightful comments to improve the current version of our manuscript. We have been able to incorporate changes to reflect all of the suggestions provided by the reviewers. We have highlighted the changes within the manuscript.

Here is a point-by-point response to the reviewers' comments and concerns.

Reviewer #2 (Remarks to the Author):

Thank you the authors for addressing previous comments and making changes to the manuscript. They have clarified some important points of earlier confusion and I believe the manuscript to have improved from the previous version. Upon rereading the new manuscript I do however remain with some comments.

Regarding the manuscript itself.

Major comments

1. After rereading the model description and the author response to an earlier comment I am now confused whether the time spent in each compartment is indeed fixed. Don't the authors model the flow between compartments as a system of ODE? In which case, the time in a compartment would be exponentially distributed. Furthermore, the authors explain they sample from parameter ranges 100 times, but is the sampling done uniformly over the range? I believe that at least for some parameters more information is known in literature about appropriate distributions for the parameters.

Authors' response: As we are solving a system of ODEs with an optimized parameter set, the time spent in each compartment is indeed fixed and therefore exponentially distributed.

When we did the perturbation analysis, we perturbed the behavioral parameters 10% of their optimized value and sampled uniformly within this range (*Parameterization>>Perturbation and parameter identifiability, line n.151-154*). As the behavioral parameters reflect the public behavior or sentiment which evolved with time, they depend upon several factors and are region-specific. Therefore, it is difficult to infer a proper distribution of such parameters. In an optimistic situation, even when such distributions can be successfully estimated for one region, extending them to other regions may be inappropriate due to the nature of events that unfolded locally. But it is true that the precise parameter distribution would improve the quality of the model. We have now discussed this in the **limitation section (line n. 617-621)**.

2. As I understand, the authors consider a random fraction of undetected cases to become detected through testing and subsequent isolation of cases. If I have understood this correctly, then the usage of the terms contact tracing or TTI are not suitable. TTI or contact tracing are generally understood to involve the explicit tracing of contacts of an index case. I don't see how explicit tracing can be investigated when the model does not keep track on who contacted whom.

Authors' response: As we are solving a system of ODEs and do not explicitly consider agents into the model, the reviewer is right that we can not track who contacted whom. We have now renamed it as *Increased Test and Isolate* (TI) strategy. In our simulation, we considered a high success rate of detection and isolation. Many countries adopted a TTI strategy that has a high success rate of detection and isolation, and several modeling studies indicated that a high proportion of cases would need to be isolated to control the pandemic. Therefore our *in silico* TI strategy is a phenomenological description of a TTI strategy. We have adapted the Results section accordingly, *Increased Test and Isolate (TI) strategy to reduce hospitalizations*, **line n.393-402**.

3. Page 11 lines 316-317: what do the authors mean by the herd immunity threshold of 70%? How do the authors go from the IFR, which they define as the proportion of deaths among infections, to making a statement about immunity on the population level? Moreover, there should at least be some references to the literature on their quoted herd immunity.

Authors' response: We thank reviewer for pointing out to this possible misunderstanding. Please note that we do not calculate herd immunity in the population from the IFR. Besides IFR, we estimated the IR (infection rate or attack rate), which corresponds to the fraction of the infected population in every region (see Table 2). We have now mentioned this in **line n. 309**. A detailed description of the estimation procedure is outlined in the supplementary material. Just to mention here that our estimated total infection is close to the Imperial college London estimation (Report 20 on Italy) and we cited this in the main text.

The herd immunity threshold can be calculated from the basic reproduction number with the formula: $1-1/R_0$. The lowest mean of reported R_0 for Covid-19 in the published articles is ~ 3 (Alimohamadi et al., 2020, doi: 10.3961/jpmph.20.076) and by that the herd immunity threshold is $\sim 70\%$. We have now summarized our calculation of herd immunity threshold to clarify this point and cited the relevant literature as requested in **line n. 315-322**.

4. I appreciate that the authors try to put their work using a compartmental model in a larger context. However, I believe that they should put more work in putting the context to using compartmental models in the COVID-19 pandemic besides the studies that are cited for Italy specifically. Moreover, I would suggest for the authors to rephrase their text: any research question require a specific model with suitable compartments. Rather they could put the emphasis on their specific research question, for which they developed the tailor-made model.

Authors' response: We modified the text as suggested and cited other studies (**line 67**).

In the *Background* section, **line n.52-60**, we acknowledged the benefits of the existing SIR and SEIR type models and we justified the need of a different model with additional compartments to study the impact of influential factors, which is not incorporated in the classical SIR or SEIR type models. In the next paragraph, we additionally justified why Covid-19 is a special case and why we need a covid specific model. We further described in the next paragraph what questions we will address with the new model.

Minor comments:

5. Please provide more detail in the references to the ISTAT data: could the authors provide a reference to the urls that they have used and the dates at which they have accessed the pages.

Authors' response: Thank you for pointing this out. This was a mistake that occurred in the bibliography compilation. We now included the urls and the info.

6. What are the consequences of assuming a fixed incubation period? Have the authors performed a sensitivity analysis.

Authors' response: The incubation period has two components, the non-infectious latent period ($1/R_2$) and the infectious subclinical period ($1/R_3$). Intuitively, in the case of a predominant latent phase, the time spent in the exposed compartment will be longer, which reduces the infectious time period. In the case of a predominant subclinical phase, the time spent in the carrier compartment will be longer, which prolongs the infectious time period. In any case, the infectious population ($CI+CR+IX+\beta*(IH+IR)$) is scaled by R_1 , which we estimated based on the daily cases. Therefore the conclusions will remain unchanged in case of an alternate incubation period.

Keeping the incubation period fixed, we performed a sensitivity analysis of R_3 upon the Reproduction number within the range provided in Table 1, and we found that the Reproduction number is less sensitive towards the variations of R_3 . We have now mentioned this in the limitation section (**line n. 603-607**).

7. Could the authors report on the number of data points in the first two weeks, to which they fit their model?

Authors' response: Daily data started from 24th February 2020, and we considered up to 8th March as the nationwide lockdown was imposed on 9th March. For each day, we considered infection, hospital, ICU and dead data. Therefore a total of 56 data points (14 daily data for four observables) were used to fit the model. We have now included this information in the *parameterization* section (**line n.127**).

8. Could the authors comment on the moving time window of seven days? Is this chosen because of the calendar week or data streams?

Authors' response: We considered a moving window with the size of a calendar week mainly for weekend correction and to avoid the daily fluctuations in the data, which depends upon the daily tests. Usually, on the weekends a lower number of tests were carried out, and therefore a bulk volume is reported either on Monday or Tuesday. We have now included it in the *parameterization* section (**line n.147**).

9. In practice, there are often delays in detection and isolation of cases, which are generally important factors in the effectiveness of these interventions. It would be good for the authors to discuss these in the limitations of their model.

Authors' response: Thanks to the reviewer for this comment. We have now discussed this in the limitation section (**line n.632**).

Thank you for the authors for pointing out the link to the github page containing the model code.

We thank the reviewer for looking at the code and for his comments that improved its readability.

code link: <https://github.com/arnabbandyopadhyay/COVID-19-in-Italy>

Regarding the code:

10. I have been unable to find a demo of the code as described in the software statement

Authors' response: We now have included a *Demo* folder that contains the first two months of Italy data. It takes ~15 min to execute the *Reference* model or *Asymptomatic* model on the dataset provided in the *Demo* folder. By following the instructions given in the *Reference_Asymptomatic* folder, part of the Fig 4 (in the main text) can be generated.

The full dataset used in this study is provided in the *Data* folder in both *Reference_Asymptomatic* and *Capacity* folder. The whole dataset or part of it can be used as a demo dataset. Instructions are given in the readme file either in the *Reference_Asymptomatic* folder or in the *Capacity* folder to run the respective version of the model.

11. It would be good to fix the formatting of the README in both PARALLEL/Reference_Asymptomatic and PARALLEL/capacity: save the README as a markdown file and format the text so that it is easier to read

Authors' response: We modified the readme file as requested.

12. What is the expected run time (and on what type of compute infrastructure, e.g. personal laptop), please report as also described in the software statement

Authors' response: The expected runtime is approximately one hour for a single region and to evaluate the complete dataset (on Intel Xeon processor @ 3.30GHz and in MATLAB 2019b). The code is parallelized in MATLAB and therefore, depending upon the availability of cores, multiple regions can be evaluated simultaneously. We have now mentioned this in the readme file.

13. Could you elaborate on the data files that are used in the code? Where and when (i.e. which files/functions) are the data being used? Furthermore, what are the minimal requirements for the data, say someone would like to run the model on their own data, is that possible under the current implementation, what are the requirements for e.g. column names. These questions should ideally be elaborated upon in a demo or README.

Authors' response: We have now enclosed this information in the readme file (>>organization/important to keep in mind section). *Models* folder contains model files (.def) necessary to evaluate the model in the Data2Dynamics framework. Model files are written in

the preferred format of Data2Dynamics; a link to Data2Dynamics is also provided in the readme file. Model equations and observables are declared in the model file. The *Data* folder contains data with column names as observables declared in the model file. The full dataset used in this study is provided in the data folder in both *Reference_Asymptomatic* and *Capacity* folders. Additional instructions are provided in the readme file to run the framework on the partial and full datasets.

To run the model on other dataset, format of data files need to be maintained and column names should be as declared in the model file (observables).

We now have included a *Demo* folder that contains the first two months of Italy data. It takes ~15 min to execute the *Reference* model or *Asymptomatic* model on the dataset provided in the *Demo* folder. By following the instructions given in the *Reference_Asymptomatic* folder, part of the Fig 4 (in the main text) can be generated.

REVIEWERS' COMMENTS:

Reviewer #2 (Remarks to the Author):

I have read the response of the authors and the revised manuscript. The authors have taken care to address all comments. Overall, the manuscript is improved from the initial submission.